# Microplastics’ Impact on the Development of AOM/DSS-Induced Colitis-Associated Colorectal Cancer in Mice

**DOI:** 10.3390/ijms262311511

**Published:** 2025-11-27

**Authors:** Natalia Zolotova, Maria Kirillova, Dzhuliia Dzhalilova, Ivan Tsvetkov, Nikolai Fokichev, Olga Makarova

**Affiliations:** 1Department of Immunomorphology of Inflammation, Avtsyn Research Institute of Human Morphology of Federal State Budgetary Scientific Institution “Petrovsky National Research Centre of Surgery”, Moscow 117418, Russia; marusyasilina99@yandex.ru (M.K.); juliajal93@mail.ru (D.D.); davedm66@gmail.com (I.T.);; 2Department of Histology, Petrovsky Medical University, Moscow 119435, Russia; 3Faculty of Biology and Biotechnology, National Research University Higher School of Economics, Moscow 109028, Russia

**Keywords:** microplastics, colitis-associated colorectal cancer, tumor nodules, inflammation, goblet cells, enteroendocrine cells, mucins, claudins, cytokines

## Abstract

Recently, evidence indicating that microplastics (MPs) have a hazardous effect on human health is accumulating. The potential of MPs having a role in carcinogenesis was suggested. Therefore, the aim of this study was to evaluate MPs’ effect on colitis-associated colorectal cancer (CAC) development. The AOM/DSS-induced CAC model was reproduced in two groups of adult male C56BL/6 mice. One of these groups received MPs (5 μm polystyrene microbeads) with drinking water at a dose of 1.48 mg/kg/day throughout the experiment (12 weeks), and the other received water untreated with MPs. In the colons of mice that consumed MPs, there was a higher number of tumor nodules at the macroscopic level, a greater tumor prevalence on histological sections, more pronounced inflammatory infiltration, a higher goblet cell volume fraction, the content of highly sulfated mucins was found in them, and there were more tumors with increased enteroendocrine cell content. We did not find any MP effects on the claudins, mucins, proapoptotic factor *Bax*, or on the proliferation marker *Mki67* gene expression in the medial colon, nor on the serum level of TNFα, IL-1β, IL-6, and IL-10 cytokines. Thus, MPs promote the CAC development in mice by exacerbating intestinal local inflammation and damaging the epithelial barrier, and MPs may represent a potential environmental cofactor contributing to CAC risk.

## 1. Introduction

Plastics are a diverse group of synthetic or semi-synthetic materials that have a polymer structure. They are characterized by their ability to be molded, extruded, or pressed into solid forms. Plastic industrial production began around the 1950s, and it is rising steadily, with projections indicating it will continue to rise significantly through to 2050. Only about 9% of all plastic waste is being recycled, and 19% is being incinerated, with the remainder ending up in landfills (50%) or mismanaged waste systems that pollute the environment (22%). Plastic pollution is a global environmental problem. The greatest concern is caused by plastic particles smaller than 5 mm—microplastics (MPs). Sometimes plastic particles smaller than 100 nm (or 1 µm) in size are distinguished as a separate subset—nano-plastics (NPs). MPs are detected in cosmetics, toothpastes, household chemicals (primary MPs), or are being formed through the aging and fragmentation of large plastic products (secondary MPs). MPs are categorized by their minute size, low mass, and environmental persistence. These properties allow MPs to spread by the wind and accumulate in soil and water, which leads to increased habitat pollution. MP particles have been found in the urban air, agricultural soils, rivers, oceans, drinking water, and human food. Therefore, the MPs’ impact on human health is an important issue [1,2,3].

Most human MP consumption evaluations are indirect and based on MP content in water and food. According to various evaluations, human MP intake is from 0.2 to 10.2 mg/kg/day [4,5] or from 1.5 to 1.1 × 10^6^ particles/kg/day [6,7,8,9,10].

MPs have been detected in many human organs, including the brain, lungs, liver, heart, spleen, testicles, bone marrow, and placenta, as well as in body fluids such as blood, urine, breast milk, semen, saliva, and follicular fluid [11,12,13]. According to numerous experimental studies, MPs can cause oxidative stress, proinflammatory reaction activation, changes in metabolism, and genotoxic effects, leading to the disruption of various organs’ structure and function [1,14,15]. Some evidence suggests a connection between MP exposure and disorders such as cardiovascular diseases, diabetes, inflammatory bowel disease (IBD), infertility, and cancer [16,17].

MPs enter the human body predominantly through ingestion with water and food. The greatest MP accumulation is observed in the large intestine. Therefore, the colon is the primary target of MP exposure [18]. Experimental studies demonstrated that MP consumption leads to oxidative stress, increased intestinal permeability, mucosa infiltration by immune cells, increased production of proinflammatory cytokines, decreased goblet cell numbers, and mucus production in the mouse colon. MPs also cause changes in proliferation, apoptosis, and differentiation of epithelial cells, expression of tight junction and glycocalyx components, membrane transport, intracellular signaling pathways, metabolome, and intestinal microflora composition [19,20]. Although exposure to MPs did not induce observable pathological alterations in the colon [19], we assume that MP-induced oxidative stress, local proinflammatory reactions, and barrier dysfunction may exacerbate the intestinal inflammation caused by other factors.

Yan et al. [21] discovered that the MP feces content in patients with IBD was higher than that of healthy individuals, and there was a positive correlation between the MP concentration in feces and the IBD severity. According to our previous studies [22,23] and the other authors’ works [24,25,26,27,28], MP consumption leads to a more severe acute and chronic colitis induced by dextran sulfate sodium salt (DSS) in mice.

IBDs, including Crohn’s disease and ulcerative colitis, are chronic, relapsing bowel disorders. Their etiology remains unclear, making prevention, diagnosis, and treatment methods ineffective. IBD is associated with a number of complications, the most severe of which is colitis-associated colorectal cancer (CAC) [29,30,31]. According to GLOBOCAN 2022, colorectal cancer (CRC) took 3rd place in incidence (9.6% of new cancer cases) and 2nd place in mortality (9.3% of cancer deaths) among malignant neoplasms worldwide [32].

Chronic inflammation in IBD leads to microenvironment development, enriched with immune cells that produce proinflammatory cytokines and growth factors, and simultaneously increases reactive oxygen species local levels that cause oxidative stress [31]. Under such conditions, the DNA damage risk increases significantly, leading to tumor-promoting gene activation and tumor-suppressor gene inactivation. Unlike sporadic CRC, in which dysplastic changes develop as discrete small lesions, in IBD, extensive areas of chronically inflamed mucosa are prone to neoplastic transformation. This phenomenon is called “field cancerization” [29,31,33]. Meta-analysis of 31,287 patients with ulcerative colitis demonstrated the overall CRC prevalence of 0.85% with a cumulative risk of 0.02%, 4.8%, and 13.9% after 10, 20, and 30 years of the disease course [34].

In recent years, the incidence of CRC in people under 50 years old has been increasing in many countries [35,36]. The reasons for this increase remain unknown, but plausible hypotheses include greater exposure to potential risk factors, such as a Western-style diet, obesity, physical inactivity, and antibiotic use. MP pollution may also play a role in the increase in CRC incidence. Various chemical compounds (PP, PE, PA, PC, PET, PS, PVC, PU, PMMA, etc.), shapes (fiber, fragment, and film), and sizes (1 µm–1.6 mm) of MP particles were detected in the tissue of human colorectal tumors [37,38,39,40,41,42,43]. Cetin et al. [42] demonstrated that in patients with colon adenocarcinoma, the number of MP particles detected in tumor tissue (702.68 ± 504.26 particles per 1 g sample) was higher than in non-tumor colon areas of the same patients (207.78 ± 154.12 particles/g) and then in patients without CRC (218.28 ± 213.05 particles/g). According to Xu et al. [44], the mean fecal MP content in CRC patients (62 particles/g) was higher than in healthy individuals (43 particles/g), and high fecal MP levels were significantly associated with an increased CRC risk.

Only five experimental studies on the MP effect on the in vivo development of CRC were published in the PubMed database up to October 2025 [39,40,45,46,47]. In an allograft CRC model in mice, it was demonstrated that a PE particle size of 500 nm modifiesthe intestinal microenvironment, thereby modulating the adaptive immune response, and promotes colorectal tumor growth in situ [47]. In models of subcutaneously implanted human colorectal carcinoma cells HCT116 and LoVo, the polystyrene (PS) particle size of 60–80 nm consumption stimulated the tumor nodules’ growth and reduced the effectiveness of antitumor therapy with oxaliplatin [39,40]. In the AOM/DSS-induced CAC model (azoxymethane and dextran sulfate sodium salt-induced colitis-associated colorectal cancer), a PS particle size of 20 nm and 60–80 nm caused an increase in the number and size of tumor nodules in the colon of mice [40,46]. However, Kuai et al. [45] reported a smaller tumor nodule size in mice with AOM/DSS-induced CAC treated with a PS size of 100 nm in comparison to animals not treated with MPs. Thus, MPs may contribute to the development of CRC, but the available studies concerning this point are few and contradictory. All of these studies included plastic nanoparticles. It was assumed that NPs are potentially more dangerous than MPs. Due to their smaller size, nanoparticles can easily cross biological barriers, such as the gut–blood barrier, and translocate to organs and tissues throughout the body. Moreover, nanoparticles can penetrate into cells, disrupting intracellular functions. NPs have a much higher surface area-to-volume ratio in comparison to MPs. This larger surface area leads to greater adsorption of harmful substances and bacteria, and to more intense release of toxic plastic components [48,49]. However, experimental data on the harm ratio of MPs and NPs is controversial [48].

The aim of our study is to assess the impact of MP consumption on CAC development in mice.

## 2. Results

The AOM/DSS-induced CAC model was replicated in two groups of mice. The “+MP” group received MPs throughout the experiment: 5 μm diameter PS particles were added to their water bowls at a concentration of 10 mg/L (consumption dose 1.48 mg/kg/day). The “–MP” group was not exposed to MPs (Figure 1).

Animals from both experimental groups had colonic mucosa macroscopic examinations that revealed multiple tumor nodules up to 4 mm in diameter, predominantly in the distal region. Tumor nodules were less common in the medial region and absent in the proximal region (Figure 2 and Appendix A). The tumor nodule number and their total area were significantly higher in animals receiving MPs. Node size did not differ between groups (Table 1). Colon length did not differ between groups and was 6.2 ± 0.5 cm (mean ± SD).

For distal colon longitudinal sections, microscopic examination of animals in both groups revealed a mosaic pattern (Figure 3). Adenocarcinomas were detected in tumors with exophytic growth without submucosa invasion, represented by multiple glands lined with proliferating atypical epithelium; crypt abscesses were detected in some crypts. Epithelialized ulcers extending to the muscularis mucosa were also observed. The ulcer surface was lined with cuboidal epithelium, while the base was composed of granulation tissue containing lymphocytes, plasma cells, macrophages, fibroblasts, and fibrocytes. There were areas with pronounced inflammatory infiltration of lymphocytes, plasma cells, macrophages, and some neutrophiles, primarily in the basal portion of the colon. Areas without pathological changes were also observed.

The prevalence of all pathological changes (tumors, ulcers, and inflammatory infiltration) and the prevalence of tumors in the distal colon were higher in animals that received MPs (Figure 4, Table 2).

In animals of both groups, the cell number in the lamina propria was significantly higher in tumor tissue than in non-tumor tissue. MP consumption led to an increase in the cell number in the lamina propria in non-tumor tissue and a tendency to increase in the tumor area (Figure 5A–D and Figure 6A). The volume fraction of PAS-positive goblet cells in tumor tissue was significantly lower than in non-tumor tissue in animals of both groups. MP consumption tended to increase the volume fraction of goblet cells in both tumor and non-tumor tissue (Figure 5E–H and Figure 6B). The content of neutral mucins in goblet cells did not differ between tumor and non-tumor tissue and was not affected by MP exposure (Figure 5E–H and Figure 6C). In animals that did not receive the MPs, there was a tendency for the content of highly sulfated mucins in goblet cells in the tumor area to decrease compared to non-tumor tissue. MP exposure increased the content of highly sulfated mucins in both tumor and non-tumor tissue (Figure 5I–L and Figure 6D).

The number of chromogranin A-positive enteroendocrine cells (EECs) in non-tumor tissue did not differ between groups and ranged from 94 to 255 chromogranin A-positive cells per mm^2^ of mucosa. Tumor nodules varied significantly in EEC content, even within the same animal. Animals not receiving MPs had tumors with both lower and higher EEC counts than non-tumor tissue. In the “+MP” group, tumors with higher EEC counts than non-tumor tissue were predominant (Figure 7).

We did not find any effect of MP consumption on the claudins *Cldn2*, *Cldn4*, and *Cldn7*, mucins *Muc1*, *Muc3*, and *Muc13*, proapoptotic factor *Bax,* and proliferation marker *Mki67* gene expression in the medial colon (Table 3), nor on the serum level of cytokines TNFα, IL-1β, IL-6, and IL-10 (Table 4).

## 3. Discussion

The number of articles concerning the MP effects on the CRC development in mice is limited. Only five previous studies described the influence on this issue [39,40,45,46,47]. The following studies used NPs: four studies used PS particles 20–100 nm in size, and one study used PE particles 500 nm in size. According to most of these studies, NPs promote tumor development. This is consistent with our data on an increase in the number of tumor nodules and the prevalence of tumors in the colon of mice with AOM/DSS-induced CAC when exposed to PS microparticles of 5 μm size. However, one study noted a reduction in tumor size due to the NPs’ effect [45]. In this study, the experimental conditions were similar to ours: we reproduced AOM/DSS-induced CAC in male C57BL/6 mice and added PS particles to the drinking bowls at a concentration of 10 mg/L. The particle size (100 nm, 50 times smaller than in our experiment), the duration of the experiment (95 days, which is 11 days longer than in our experiment), and the DSS cycles (6 days 2.5% DSS + then 15 days water vs. 7 days 1% DSS + 14 days water) differed.

According to the literature, tumors of patients with CRC demonstrate a higher number of inflammatory cells and a lower number of goblet cells than in normal tissue [50]. Furthermore, patients with CRC have a decreased production of the main structural component of mucus, mucin 2 [51]. These data are consistent with the differences we observed between tumor and non-tumor colon tissue in mice.

The impact of MPs on the severity of inflammatory infiltration and the goblet cell response in mouse models of CRC had not been previously assessed. We identified trends toward increased inflammatory infiltration and goblet cell volume in both tumor and non-tumor colon tissue in mice exposed to MPs. In healthy mice, according to numerous studies, MP consumption leads to the infiltration of the colonic mucosa by immune cells, primarily lymphocytes, increased levels and mRNA expression of proinflammatory cytokines, and oxidative stress development. MP exposure also leads to an increase in the proportion of neutrophils and proinflammatory macrophages and a decrease in the ratio of anti-inflammatory macrophages in the colon [19]. We previously showed that in healthy mice, after 6 and 12 weeks of MP consumption, the number of cells in the colonic mucosa increased. In acute and chronic colitis, MP consumption, although not affecting the density of inflammatory infiltration, resulted in the involvement of a significantly larger area of the colonic mucosa in the pathological process [22,23]. Chronic inflammation can lead to oxidative stress and DNA damage, which contribute to cancer initiation and progression. Inflammation creates a tumor microenvironment that can promote proliferation and resist cell death, facilitating tumor progression [29,31,33]. It is likely that the increase in the number of tumor nodes observed in the experimental study performed when exposed to MP is due to increased inflammatory reactions.

Goblet cells normally comprise up to 16% of colonic epithelial cells. They produce mucus, which prevents the adhesion and invasion of pathogenic microorganisms, serves as a substrate for the attachment and nutrition of commensal microflora, and also acts as a lubricant, facilitating the passage of chyme [52]. In healthy mice, according to most studies, MP consumption leads to a decrease in the number and volume fraction of goblet cells, the content of the main structural component of mucus, mucin 2, and the expression of its mRNA [19]. However, we detected a tendency toward an increased volume fraction of goblet cells in tumor and non-tumor colon tissue in mice treated with MP. In healthy animals, MP likely induces hypersecretion and depletion of goblet cells. In colitis-associated CRC, severe disruption of the epithelial barrier leads to goblet cell hypertrophy and hyperplasia. In patients with CRC, higher MUC2 expression was negatively correlated with TNM stage, lymphatic metastasis, and CRC prognosis [51].

Mucin terminal glycans can be modified by sulfation or sialylation, designating them as acidic mucins; those without these changes are neutral mucins. The acidic forms, especially sulfomucins, are more resistant to enzymatic degradation by bacterial and host-derived enzymes than their neutral counterparts [53,54]. Published research presents contradictory findings concerning how MP exposure affects the balance of acidic versus neutral mucin levels in healthy mouse goblet cells [19]. We investigated an increase in the levels of highly sulfated mucins in goblet cells in tumor and non-tumor tissue in mice with CAC exposed to MPs, which likely represents a protective response to disruption of the epithelial barrier.

EECs are cells of the gastrointestinal tract that secrete hormones that regulate gastrointestinal motility, nutrient sensing, and glucose homeostasis. They also play a role in immune responses and can influence appetite and satiety. In the colon, EECs comprise approximately 1% of epithelial cells. Chromogranin A is a general marker for EECs [55]. According to single-cell transcriptome analysis [56], in the mouse colon EEC are represented by the following four cell types: (1) 48% Enterochromaffin (EC) cells, which secrete serotonin (5-TH); (2) 35% L-cells, which secrete Glucagon-like peptide-1 and Peptide YY; (3) 11% D-cells, which secrete Somatostatin; and (4) 7% Insulin-like peptide 5 (Insl5)-producing cells. Thus, the predominant hormone of the colon is serotonin. In the intestine, serotonin stimulates peristalsis, mucus secretion, chloride, and bicarbonate, inhibits water absorption, causes vasodilation, and modulates immune responses by attracting immune cells and exerting a proinflammatory effect [57]. Our previous studies demonstrated that in healthy mice exposed to MPs, the number of EECs and immune cells in the colon increases and the goblet cell volume fraction decreases [22,23]. We suggest that MPs mechanically irritate the colonic wall, causing increased secretion and production of serotonin. Serotonin induces hypersecretion and the emptying of colonic cells, and moreover, it attracts immune cells to the intestinal mucosa. These are adaptive responses aimed at flushing MPs from the body and strengthening local immune defenses. In mice with CAC, the number of EECs in non-tumor tissue did not change when exposed to MP, but in tumors, it varied significantly in both MP-exposed and non-MP-exposed mice. However, in mice consuming MPs, the number of tumor nodules with increased EEC content was higher. It could be suggested that EEC hormones may modulate tumor progression by stimulating epithelial cell proliferation and vascularization, and by recruiting immune cells. However, research on the connection between the presence of enterocytes in colorectal carcinomas and prognosis is inconclusive [58]. According to Chen et al. [59], a high EEC number and chromogranin A level in CRC are positively correlated with invasion and lymph node metastasis.

In addition to secretory mucin 2, which forms the mucus layer, the colon expresses a number of transmembrane mucins that are part of the glycocalyx, a component of the intestinal protective barrier, and are also involved in cell signaling. MUC1 is typically expressed at low levels in normal colon tissue but is overexpressed in a high percentage of CRC cases. Its expression level often increases with advancing tumor stage and metastasis. MUC13 was also found to be upregulated in primary and metastatic CRC. Its overexpression is associated with resistance to chemotherapy [60]. The human colonic glycocalyx is primarily composed of MUC3, MUC12, and MUC17 mucins. These proteins share significant structural similarities, a finding explained by their corresponding genes being orthologous to the single *Muc3* gene found in mice [61]. In CRC, MUC3 is often overexpressed, and high MUC3 levels are associated with a poor cancer prognosis. The expression of MUC12 in CRC decreases, and low MUC12 expression is associated with a lower survival rate [60]. In CRC, the transmembrane mucin MUC17 is typically downregulated, or its expression is lost entirely [62].

Tight junctions in the colon are protein complexes that create a selective barrier between adjacent epithelial cells. They regulate the movement of ions, water, and other molecules through the intercellular space. Claudins are integral tight junction proteins. In CRC, claudin-2 (CLDN2) is typically overexpressed and promotes the aggressive characteristics of cancer cells, including proliferation, metastasis, and chemoresistance. High CLDN2 expression is also associated with poor patient prognosis [63]. Claudin-4 (CLDN4) is often overexpressed in primary CRC and metastatic sites but downregulated in invasive cancer cells [64]. Claudin-7 (CLDN7) expression is frequently downregulated in CRC tissues in comparison to a normal colon, and this loss is associated with aggressive tumor features, increased metastasis, and poor patient prognosis [65].

In cancer, the normal balance between cell proliferation and apoptosis is disrupted. An uncontrolled increase in cell proliferation combined with an inhibition of apoptosis allows cancer cells to accumulate and develop the tumor. Ki-67 (encoded by the *MKI67* gene) is a protein that serves as a biomarker for cell proliferation. Bax is a key proapoptotic protein [66,67].

We did not detect the effect of MPs on the mucins, claudins, Mki67, and *Bax* gene expression in the medial colon of mice with CAC. In our previous study, in healthy mice consuming MPs for 12 weeks, we did not detect changes in the expression of *Muc1*, *Muc3*, *Muc13*, *Cldn4*, *Cldn7*, *Mki67*, and *Bax* mRNA, and noted a tendency for *Cldn2* to increase. We probably did not detect an MP effect on gene expression, since the effects of MPs are too weak compared to the effects of AOM/DSS. However, it is also possible that we unsuccessfully selected samples for analysis. We collected the medial colon for qRT-PCR analysis, although most tumor nodules were detected in the distal colon. In addition, we collected the entire colon, regardless of the presence and number of tumors in it, and according to the study of Dzhalilova et al. [68], gene expression levels in the tumor nodule, peritumoral zone, and normal colon tissue in mice with CAC differ significantly. Future similar studies should separately collect tissue from the tumor nodule and peritumoral zone, preferably from the distal colon.

In addition to local symptoms, CRC also causes a range of systemic manifestations, including a systemic inflammatory response. This chronic, dysregulated inflammatory state significantly impacts tumor progression and is a marker of poor prognosis [69]. In comparison to the healed individuals, CRC patients demonstrated from 3.2- to 4.4-fold higher concentrations of the proinflammatory cytokines, including IL-1β, IL-6, and TNF-α. On the contrary, the level of the anti-inflammatory cytokine IL-10 was reduced [70]. Comparing the cytokine levels in this study with our previous analysis of cytokine levels in healthy animals [23], both MP-treated and MP-untreated mice with CAC had approximately 1.5-fold elevated TNF-alpha levels, while other cytokine levels were normal. We revealed no effect of MPs on serum levels of IL-1β, IL-6, TNF-α, and IL-10 in mice with CAC. It is likely that the MP effects are too weak in the context of the severe pathological process caused by AOM/DSS. The exact mechanism by which DSS causes colitis is not fully understood. It was demonstrated that DSS disrupted the mucus layer, caused epithelial cell death, and stimulated the innate immune response, triggering the secretion of proinflammatory cytokines and chemokines. Damage to the epithelial barrier leads to the translocation of dietary and bacterial antigens to the mucosa and the development of an inflammatory reaction in the intestinal wall. Increased intestinal barrier permeability, especially in ulcer areas, led to the release of luminal antigens into the bloodstream, which triggers systemic inflammatory reactions [71]. MPs also caused an increase in colon permeability: a number of studies demonstrated the increase in intestinal permeability for fluorochrome-labeled dextran with a molecular weight of 4 kDa and 70 kDa when consuming MPs [18,72,73,74,75,76,77]. But the effects of MP are much weaker than those of DSS.

### Limitations of the Study and Further Research Objectives

Experimental sample size. Data from the literature on the effect of MPs on tumor development are still quite sporadic and contradictory. The experimental group sizes in published articles are small: from 4 to 10 mice per group [39,40,45,46]. Therefore, we used a small number of experimental animals (*n* = 7). According to the power analysis (two independent study groups, continuous, effect size mean, enrollment ratio = 1, type I/II-error rate, Alpha = 0.05, and Power = 80%), sample size *n* = 7 is sufficient for parameters such as tumor node number and total area of tumor nodes on macrophotos, all pathological changes prevalent in histological sections, lamina propria cell number in tumor and non-tumor tissue, and goblet cell highly sulfated mucin content in non-tumor tissue. For tumor prevalence and goblet cell highly sulfated mucin content in tumor tissue, *n* = 8 mice are required, and for goblet cell volume fraction and neutral mucin content, 11–18 animals per group are required (Appendix A).

MP particles shape, size, and type. MP particles detected in human colorectal tumors vary considerably according to chemical structure (PP, PE, PA, PC, PET, PS, PVC, PU, PMMA, etc.), shape (fiber, fragment, and film), and size (1 µm–1.6 mm) [37,38,39,40,41,42,43]. However, experimental conditions must be standardized, controlled, and reproducible. Therefore, experimental models of MP exposure generally use only one type of spherical microparticle of a single standard size for each experiment [39,40,45,46,47]. We also used one type of particle: 5 µm polystyrene microspheres. However, there is evidence that the effects of magnetic fields depend on the size, shape, type of particles, and their charge. The ratio of immune cells in the colon mucosa depends on the type of MP: PS > PVC > PET > PE > PP > control [78]. Modified nanoparticles PS-NH2 and PS-COOH more easily enter Caco-2 cells and demonstrate stronger toxicity in mice in comparison to nonmodified PS [79]. According to some of the obtained data, smaller particles can easily cross the intestinal barrier, accumulate in larger quantities in internal organs, and cause more pronounced damage [80,81], but according to other data [82], PS particles of 5 μm in size cause more pronounced proinflammatory changes in the colon of mice than particles of 0.2 and 1 μm in diameter. Studies where a mixture of different MPs was used are rare [18,83]. When two different-sized particles are used together, their bioaccumulation changes [18]. The effects of different MPs on the development of CRC should be studied in the future.

Physicochemical characteristics of MP. We did not study the MP particles themselves or their properties in suspension. We based our study on the only information provided in the product description on the supplier’s website: Quality level of 100, form aqueous suspension, crosslinking was 0% cross-linked, concentration was 10% (solids), particle size was 5 μm std dev < 0.1 μm, coefficient var < 2%, density was 1.05 g/cm^3^, and storage temp 2–8 °C. However, as noted above, the effects of MPs depend on the particle’s properties.

MP intake dose. There are two main methods for modeling oral MP intake: (1) administering MPs via a stomach tube; and (2) adding MPs to drinking bowls [1]. In the first case, the dose can be precisely controlled. However, humans are continuously exposed to MPs through water and food throughout the day. With intragastric administration, mice receive the entire daily dose at one point: the peak dose of MPs is very high, followed by no further MP intake for the next 24 h. Furthermore, the effects of MPs on healthy mice are relatively weak in comparison to nonspecific stress reactions [19,84]. Daily insertion of a tube into the stomach is a strong stressor for mice. The effects of the procedure itself may overwhelm the effects of MPs. In the second option, when adding MPs to the water bowls, it is technically impossible to control the exact liquid volume consumed by each mouse. In addition, the suspension of MP particles is unstable; the particles gradually settle, but the sedimentation rate is slow. Calculated using Stokes’ Law, the settling velocity for 5 µm spherical polystyrene particles in distilled water is 7.03 × 10^−7^ m/s, or approximately 6 cm per day. Although this model does not allow for controlling MP doses precisely, it is a significantly better match with the dynamics of human MP consumption and does not provide stress for animals.

MP bioaccumulation. We did not evaluate MP accumulation in mouse tissues and feces. However, other authors conducted bioaccumulation studies on the MPs we used (5-μm polystyrene microspheres). According to Liang et al. [18], after 24 h exposure, MPs predominantly accumulated in the mice’s colons. Deng et al. [85] revealed that the particle concentration in the liver, kidney, and gut tissues reached an optimum state within 14 days of the exposure onset, and after a 4-week exposure, MP colon concentration was 1.4 mg/g. In mice with acute colitis, MP accumulation in the abdomen significantly increased compared to healthy animals [25].

Colon tissue sampling. Pathological changes in the colon in the AOM/DSS-induced CAC model were unevenly distributed. Most tumor nodules were detected in the distal colon, so it is necessary to collect material from the distal colon for analysis. Furthermore, tumor and non-tumor tissue differ significantly, so the study should be conducted separately for these tissue types. We likely did not detect the effect of MPs on gene expression in the colon because we sampled the medial colon and did not separate the tissue into tumor and non-tumor tissue. That was a significant limitation of our study. However, the mouse colon is small, making it impossible to collect material from the same colonic region simultaneously for morphometric histology and PCR analysis of gene expression, much less to additionally assess protein levels in the intestinal wall. For molecular biology studies, it is advisable to use additional, separate groups of animals.

Local markers of inflammation. We assessed the severity of inflammation using two parameters: cell density in the lamina propria and serum cytokine levels. Although we found increased immune cell infiltration into the intestinal wall following MP exposure, systemic cytokine levels did not differ. Therefore, it is worth examining the effects of MPs on local levels of pro- and anti-inflammatory cytokines in the intestinal wall at the level of mRNA expression and protein content. According to most of the literature, in healthy mice, colon MP consumption led to an increase in the proinflammatory cytokines TNFα, IL-1β, and IL-6 levels, as well as their mRNA expression [19]. In mice with acute and chronic DSS-induced colitis, after MP exposure, they demonstrated higher colon mRNA expression of proinflammatory cytokine genes (*Tnfa*, *Il1b*, *Il6*, *Il17a*, *Il22*, and *Tgfb*), lower expression of anti-inflammatory cytokine *Il10*, and increased colonic level of proinflammatory cytokines TNF-α, IL-1β, and IL-6 [26,27,28,86]. According to Dzhalilova et al. [68], in mice with AOM/DSS-induced CAC, cytokine expression in tumor and non-tumor colon tissue differs significantly, so the study must be conducted separately for different tissue types. In addition, we determined the severity of inflammatory infiltration based on the number of all nuclei in the lamina propria. In addition to inflammatory immune cells (proinflammatory macrophages, lymphocytes, plasma cells, and neutrophils), the lamina propria also contains fibroblasts, fibrocytes, anti-inflammatory macrophages, and regulatory T lymphocytes. Migrating tumor cells may also be present in the tumor area. Therefore, it is also worth assessing the effect of MPs on different immune cell population contents.

Markers of tumor malignancy and progression. All the tumors we identified exhibited an abnormal growth pattern, consisting of multiple glands lined with proliferating atypical cells with hyperchromatic nuclei of varying shapes and heights. Goblet cell number and mucus production in the tumors were significantly reduced, indicating impaired epithelial cell differentiation. The tumor stroma consisted of connective tissue infiltrated with neutrophils, lymphocytes, and histiocytes. None of the cases demonstrated tumor invasion into the submucosa. As both tissue and cellular atypia were observed, we classified the tumors we observed as adenocarcinomas. However, for an accurate assessment of the differentiation progression and malignancy of the tumor, an immunohistochemical study of a number of markers should be carried out: proliferation marker Ki-67, a component of the Wnt signaling pathway involved in the regulation of proliferation and migration—β-catenin, apoptosis inductor caspase-3, tumor-suppressor p53, and proteins of intermediate filaments of epithelial cells cytokeratins [87].

Mechanisms of MP effect. Our aim was to reveal whether MPs have an effect on colorectal tumor growth in vivo; we did not investigate the mechanisms involved. In vitro studies demonstrated that MPs induce reactive oxygen species development and oxidative DNA damage, which leads to decreased cell viability and the initiation of autophagy or apoptosis [88]. In vivo, in the healthy mouse colon, the main negative effects of MPs are increased intestinal permeability, activation of oxidative stress and proinflammatory reactions, and changes in the microbiota composition [19]. Moreover, suppression of the gut microbiota with antibiotics mitigates the effects of MPs [89,90,91,92]. According to Tian et al. [46], in mice with AOM/DSS-induced CAC, exposure to 20 nm polystyrene nanoparticles in the colon resulted in suppression of fatty acid metabolism, activation of lipid peroxidation, increased ROS levels, significantly more severe DNA damage, increased abundance of colitogenic bacterium *Allobaculum* in the intestinal microflora, and decreased postbiotic bacterium *Lactobacillus*. Further studies concerning MPs’ effect mechanisms in tumor development are an important scientific goal.

Sex differences. Sex hormones and sex chromosome genes significantly influence the immune system, leading to sex differences in the incidence and the severity of infectious, inflammatory, autoimmune, and neoplastic diseases [93,94]. In particular, CRC morbidity and mortality are higher in men than in women. Moreover, left-sided CRC is predominant in men, while women are more often diagnosed with the more aggressive right-sided CRC [95]. In the AOM/DSS-induced CAC model, the number of tumor nodules and their sizes are greater among males than among females [96]. Therefore, it is impossible to include both males and females in the same sample. The female hormonal cycle leads to cyclical changes in immune function [97]. In women, many chronic diseases experience worsening symptoms before or during menstruation [98]. In acute ozone exposure in the lungs of female mice in the follicular phase (proestrus, estrus), a significantly more pronounced proinflammatory reaction is observed than in the luteal phase (metestrus, diestrus) [99]. Therefore, studies on females must take into account the stability and phase of the estrous cycle, which makes the work more labor-intensive. Furthermore, fluctuations in sex hormones increase the variability of responses, so studies on females, especially with small sample sizes, risk missing effects. Therefore, most studies are first conducted on males, and the results are then confirmed and supplemented with sex-specific data in studies on females. Undoubtedly, future research on MPs’ effect on colorectal cancer development in females is an important task.

## 4. Materials and Methods

### 4.1. Animals

A total of 16 mature male C57BL/6 mice were sourced from the Stolbovaya branch of the Federal State Budgetary Institution of Science, Scientific Center for Biomedical Technologies of the Federal Medical and Biological Age (Chekhov, Russia). The mice were housed in groups of eight per cage within an open system maintained at room temperature, with natural light exposure. When working with experimental animals, Directive 2010/63/EU of the European Parliament and of the Council of 22 September 2010 on the protection of animals used for scientific purposes was followed.

### 4.2. Experimental Groups

Animals were divided into two groups of eight mice (Figure 1): (1) the “–MP” control group, not exposed to MPs; and (2) the “+MP” group, which consumed MPs. MP particles were added to the water bowls throughout the experiment. Both groups simulated the AOM/DSS-induced CAC model. One animal died in each group during the experiment. Animals were sacrificed on day 84 by cervical dislocation under zoletil anesthesia (Virbac Sante Animale, Carros, France) (Figure 1).

Animals were randomly allocated to groups. Technical personnel handling the mice were unaware of the experimental design. To maintain consistency, all experimental procedures occurred at the same time, and all animal housing units were situated side-by-side in the same room. The experimental unit was a single animal. Animals that died before the end of the experiment were excluded from the study. The sample size was determined according to similar studies [40,45].

### 4.3. Microplastic Consumption Model

Throughout the experiment, PS (polystyrene) particles 5 μm in size (79633, Supelco, Bellefonte, PA, USA) were added to the animals’ drinking bowls to a final concentration of 10 mg/L, which is consistent with previously published studies [1,19]. To prevent the introduction of foreign MP particles, distilled water and glass drinking bowls were used. To prevent particle settling, the drinking bowls were shaken several times a day. Twice a week, the MP suspension in the drinking bowls was completely replaced with a new one, and the volume of remaining liquid was measured. Animal weight was determined once a week. Animal weight and fluid consumption did not change in a statistically significant way during the experiment, did not differ between groups, and amounted to 23 ± 2.3 g and 3.4 ± 0.7 mL/mouse/day (Mean ± SD), respectively. The MP dose was approximately 1.48 mg/kg/day (2 × 107 particles/kg/day), which is consistent with the estimates of Senathirajah et al. [4] for human MP intake.

### 4.4. AOM/DSS-Induced Colitis-Associated Colorectal Cancer Model

On the first day of the experiment, the animals were intraperitoneally injected with the carcinogen azoxymethane (AOM) (A5486, Sigma-Aldrich, St. Louis, MO, USA) at a dose of 10 mg/kg of body weight. Starting on the 8th, 29th, and 50th days of the experiment, DSS (Dextran sulfate sodium salt, Mr ~40,000, AppliChem, Darmstadt, Germany) was added to the drinking water for 7 days for a final concentration of 10 g/L (1%) to induce inflammation [29].

### 4.5. Macroscopic Evaluation of Tumor Nodes in the Colon

The colon was flattened on a filter, opened along the mesentery, and removed of its contents. Photos were taken with a Nikon D3100 camera (Tokyo, Japan) from a distance of approximately 20 cm. The number of tumor nodules and their area were determined using imageJ software (Version 1.54i). Node diameter was calculated, assuming tumor nodules to be approximately round. Nodes with a diameter of at least 1 mm were considered.

### 4.6. Histological Study of the Colon

#### 4.6.1. Preparation of Histological Specimens

A 1.5 cm segment of the distal colon was excised and submerged in 10% neutral buffered formalin (Biovitrum, Saint Petersburg, Russia) for 24 h of fixation. The tissue was then rinsed under tap water and preserved in 70% ethanol until further use. Following standard histological processing and embedding in Histomix, 5 μm-thick longitudinal sections were prepared. The resultant sections were subjected to various staining protocols: hematoxylin and eosin (H&E), Alcian blue pH 1.0 (targeting highly sulfated mucins), a Periodic Acid-Schiff (PAS) reaction (for neutral mucins), and immunohistochemistry using antibodies against chromogranin A (for EECs).

#### 4.6.2. Prevalence of the Pathological Process

Slides stained with H&E were scanned at 100× magnification. Within the QuPath-0.5.1 software environment, measurements were taken along the muscularis mucosae to determine the length of areas free of pathology, as well as those containing inflammation, ulcers, or tumors. The percentage of the total section length presenting pathological changes was then determined.

#### 4.6.3. Severity of Inflammatory Infiltration

In the QuPath-0.5.1 software on scanned images of the H&E-stained colon sections, the mucosa area and number of cell nuclei on it were counted in regions with and without tumors. The cellular density of the lamina propria was expressed as cells per 1 µm^2^.

#### 4.6.4. Goblet Cell Content and Mucus Properties

Mucin terminal carbohydrate groups were either unmodified or acid-modified. The Periodic Acid-Schiff (PAS) reaction identifies unmodified groups: periodic acid oxidizes vicinal diols into dialdehydes, which subsequently react with the colorless Schiff’s reagent. Conversely, Alcian blue acts as a basic dye that attaches to highly sulfated mucins (strong acids) when the pH is adjusted to 1.0. Images of sections processed with both Alcian blue pH 1.0 and the PAS reaction were captured at 100× magnification in tumor and non-tumor regions. The goblet cell area and the brightness of their staining were determined. In the ImageJ software (Version 1.54i), binarization of the images was performed, setting the threshold so that only goblet cells were highlighted. The area spanning the colonic wall from the muscularis mucosae up to the lumen was manually outlined, and the total mucosal area and the included goblet cell area were determined. The volume fraction of goblet cells was calculated as a ratio of their area to the total mucosal area (PAS reaction images). Average brightness measurements were taken for both the goblet cells themselves and the image background (areas without tissue). The optical density of the goblet cells was derived by taking the logarithm of the ratio of average background brightness to average goblet cell brightness. Elevated optical density readings correlate directly with increased content of highly sulfated (Alcian blue) or neutral (PAS-reaction) mucins.

#### 4.6.5. Enteroendocrine Cell Content

EECs were detected via immunohistochemical staining using a rabbit anti-mouse polyclonal antibody targeting chromogranin A (ab15160, Abcam Inc., Cambridge, UK). A secondary antibody with a peroxidase label (Donkey anti-Rabbit IgG (H+L) Highly Cross-Adsorbed Secondary Antibody, HRP, A16035, Invitrogen, Carlsbad, CA, USA) was subsequently applied. Images of the sections were captured at 100× magnification in both tumor and non-tumor regions. The total mucosal area was measured, and the number of chromogranin A-positive cells within that area was quantified. The final calculation expresses the density of these positive cells as counts per 1 mm^2^ of the mucosal area.

### 4.7. Real-Time PCR

A 1 cm fragment from the medial colon was preserved in IntactRNA fixative (BC031, Evrogen, Moscow, Russia) for later use in RNA extraction and gene expression profiling.

qRT-PCR was employed to assess gene expression changes in the medial colon for several target genes: tight junction proteins (*Cldn2*, *Cldn4*, and *Cldn7*), glycocalyx components (*Muc1*, *Muc3*, and *Muc13*), the proapoptotic factor *Bax*, and the proliferation marker *Mki67*. RNA isolation was accomplished using the RNA Solo kit (Eurogen, Moscow, Russia), followed by cDNA synthesis performed with the MMLV RT kit (Eurogen, Moscow, Russia). Relative expression levels for all genes were normalized against the β-actin (*Actb*) mRNA expression level. The PCR amplification used a 5× qPCRmix-HS SYBR mixture (Eurogen, Moscow, Russia) and 0.2–0.4 µM final concentration oligonucleotides on a DTprime instrument (DNA-Technology, Moscow, Russia). All PCR primers were designed using the online program primer-BLAST (https://www.ncbi.nlm.nih.gov/tools/primer-blast/index.cgi, accessed on 21 October 2025), adhering to standard requirements, and were synthesized commercially by Eurogen (Moscow, Russia). Primer sequences are in Table 5.

The relative concentration of mRNA was calculated using the following Formula (1):(1)$$\frac{\;[A]0}{\;[B]0}=E^{∆C(T)}$$
where [A]0 is the initial concentration of gene mRNA in the PCR mixture; [B]0 is the initial concentration of *Actb* mRNA in the PCR mixture; E is the reaction efficiency (taken as 1.98); and ΔC(T) is the difference between the threshold cycles of *Actb* and the target gene.

### 4.8. ELISA

Blood samples were drawn from the neck vessels. The serum content of the cytokines TNF-α, IL-1β, IL-6, and IL-10 was measured using commercially available ELISA kits (FineTest, Wuhan, China).

### 4.9. Statistics

Statistical processing of all generated data was carried out using the STATISTICA 6.0 software (StatSoft, Inc., Tulsa, OK, USA). Given the small sample size and abnormal distribution patterns (*x*^2^ criterion), nonparametric methods were selected. Data description uses the median and interquartile ranges, Me (25%; 75%). The Mann–Whitney test was employed for inter-group comparisons, with Bonferroni correction applied for multiple testing. We defined statistical significance as *p* < 0.05 for comparisons between two groups and *p* < 0.0085 for four groups. *P*-values between 0.0085 and 0.05 in four-group comparisons were classified as “trends.”

## 5. Conclusions

In the colons of mice with AOM/DSS-induced CAC that consumed PS microparticles 5 μm in size at a dose of 1.48 mg/kg/day for 12 weeks, a higher number of tumor nodules at the macroscopic level and a greater tumor prevalence on histological sections were observed in comparison to animals that did not receive MPs. Animals receiving MPs demonstrated more pronounced inflammatory infiltration in both tumors and non-tumor tissue. More pronounced inflammation promotes tumor development. Furthermore, we identified the effect of MPs on epithelial barrier components: animals receiving MPs demonstrated higher goblet cell volume fraction and highly sulfated mucin content in both tumor and non-tumor tissue, and they also had more tumor nodules with an increased content of EECs. In humans with CRC, increased production of the main component of goblet cells, mucin 2, and a high tumor content of EECs are associated with a poor prognosis. Therefore, MPs promote the development of CAC in mice by exacerbating intestinal inflammation and damaging the epithelial barrier. MPs may represent a potential environmental cofactor contributing to CAC development risk. However, this issue requires further research.

## Figures and Tables

**Figure 1 ijms-26-11511-f001:**
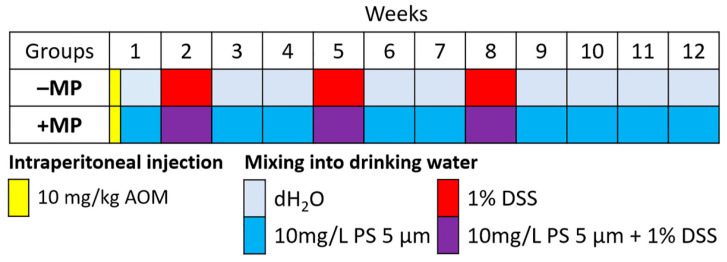
Experimental design. Groups of mice with AOM/DSS-induced colitis-associated colorectal cancer, which did not receive microplastics (–MP) and received microplastics (+MP). AOM—azoxymethane, dH_2_O—distilled water, DSS—dextran sulfate sodium salt, PS—polystyrene.

**Figure 2 ijms-26-11511-f002:**
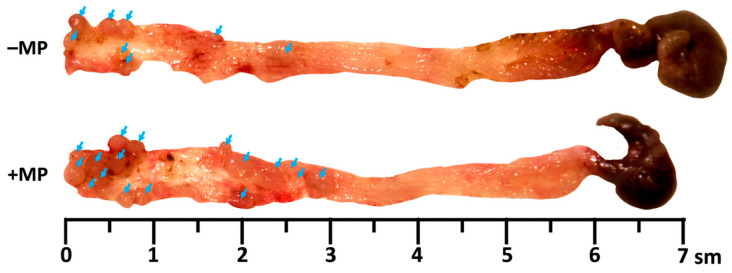
Tumor nodules (blue arrows) in the colon of mice with AOM/DSS-induced colitis-associated colorectal cancer, which did not receive microplastics (–MP) and received microplastics (+MP).

**Figure 3 ijms-26-11511-f003:**
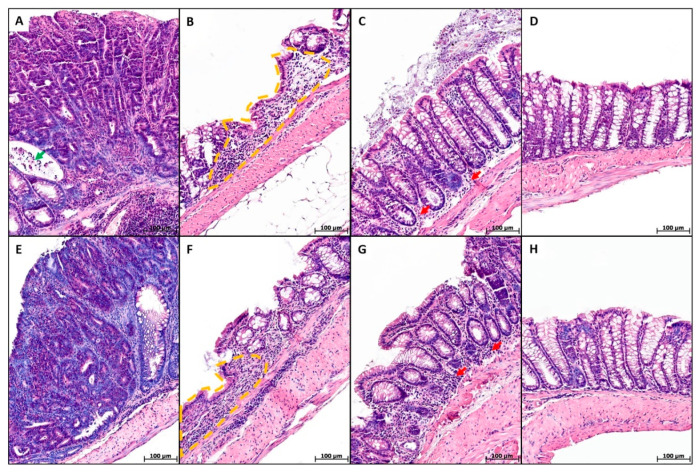
Distal colon of mice with AOM/DSS-induced colitis-associated colorectal cancer that did not receive microplastics (**A**–**D**) and received microplastics (**E**–**H**), H&E. (**A**,**E**)—adenocarcinomas (green arrow—crypt abscess), (**B**,**F**)—ulcers (orange dotted line), (**C**,**G**)—inflammatory infiltration (red arrows), and (**D**,**H**)—areas without significant pathological changes.

**Figure 4 ijms-26-11511-f004:**
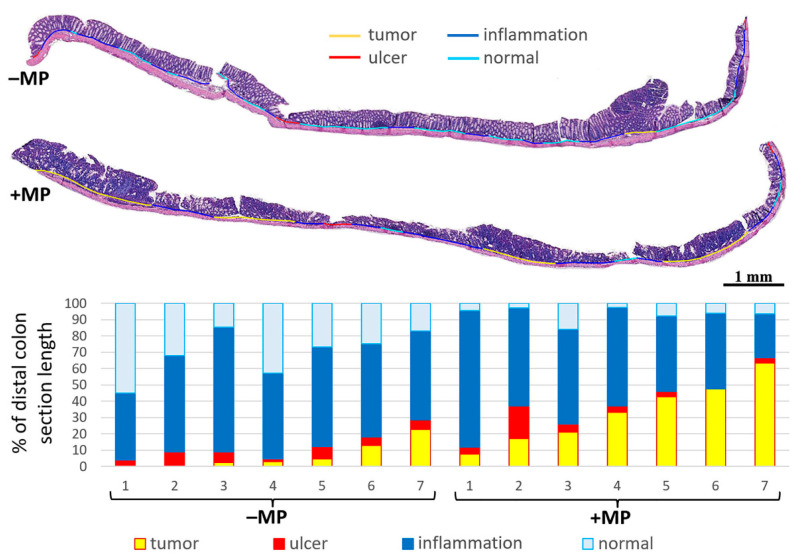
Pathological changes prevalent in the distal colon of mice with AOM/DSS-induced colitis-associated colorectal cancer that did not receive microplastics (–MP) and received microplastics (+MP).

**Figure 5 ijms-26-11511-f005:**
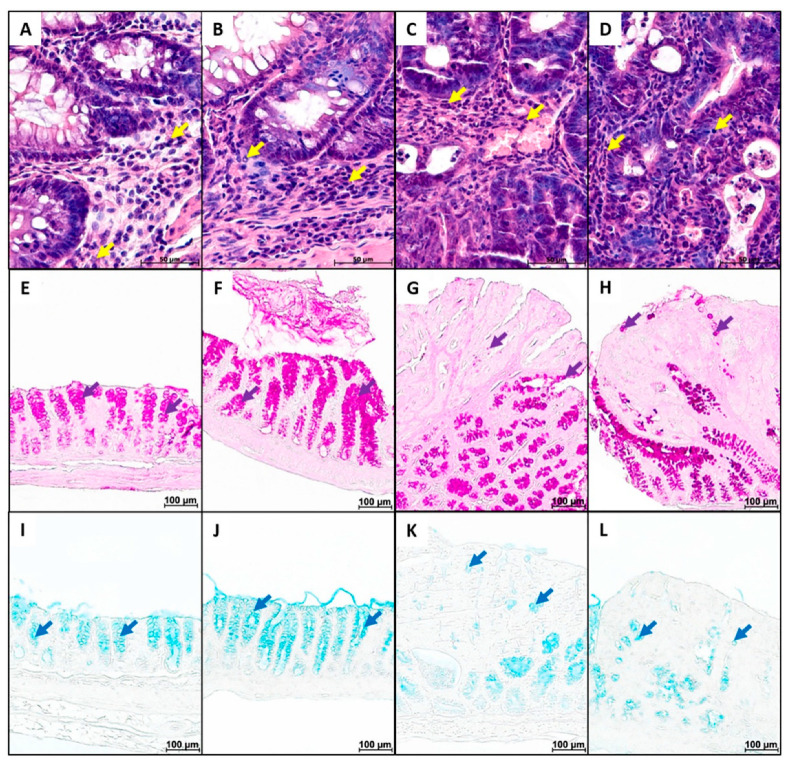
Areas of non-tumor (**A**,**B**,**E**,**F**,**I**,**J**) and tumor (**C**,**D**,**G**,**H**,**K**,**L**) tissue from the distal colon of mice with AOM/DSS-induced colitis-associated colorectal cancer that did not receive microplastics (**A**,**C**,**E**,**G**,**I**,**K**) and received microplastics (**B**,**D**,**F**,**H**,**J**,**L**). (**A**–**D**) H&E; (**E**–**H**) PAS-reaction; (**I**–**K**) Alcian blue staining. Yellow arrow—immune cell infiltration, purple arrows—neutral mucins, and blue arrows—highly sulfated mucins.

**Figure 6 ijms-26-11511-f006:**
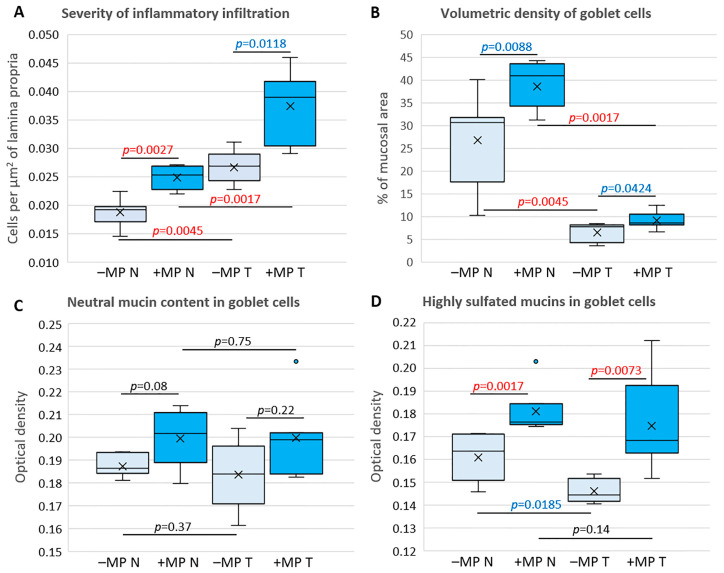
Changes in the lamina propria cell number (**A**), goblet cell volume fraction (**B**), and the content of neutral (**C**) and highly sulfated (**D**) mucins in goblet cells in non-tumor (N) and tumor (T) tissue in the distal colon of mice with AOM/DSS-induced colitis-associated colorectal cancer that did not receive microplastics (–MP) and received microplastics (+MP). Red text—statistically significant differences (*p* < 0.0085), and blue text—trends (*p* < 0.05); light boxes—“–MP” group, dark boxes—“+MP” group; dots—outliers; horizontal black lines connect pairs of compared data samples.

**Figure 7 ijms-26-11511-f007:**
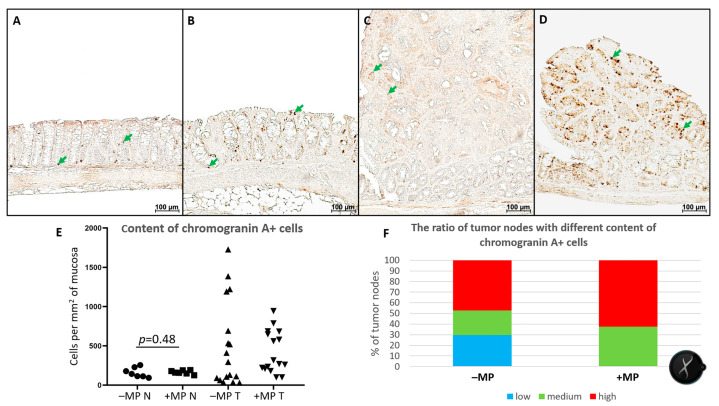
Chromogranin A^+^ enteroendocrine cells in the distal colon of mice with AOM/DSS-induced colitis-associated colorectal cancer for those that were untreated with microplastics (–MP) and those that received microplastics (+MP). (**A**) Non-tumor colon tissue of a mouse that did not receive microplastics; (**B**) non-tumor colon tissue of a mouse consuming microplastics; (**C**) tumor with low chromogranin A+ cell level; (**D**) tumor with high chromogranin A^+^ cell level; green arrow—chromogranin A^+^ enteroendocrine cells; (**E**) chromogranin A^+^ cell level in non-tumor (N) and tumor (T) tissue in mice with CAC untreated (–MP) and receiving (+MP) microplastics (for tumors, each point is one tumor nodule), horizontal black line connect pair of compared data samples; (**F**)—the ratio of the number of tumor nodes in which the chromogranin A^+^ cell level was lower (<94 cells/mm^2^—low), the same (94–255 cells/mm^2^—medium), and higher (>255 cells/mm^2^—high) than in non-tumor tissue.

**Table 1 ijms-26-11511-t001:** Tumor nodes number and size in the colon of mice with AOM/DSS-induced colitis-associated colorectal cancer, which did not receive microplastic (–MP) and received microplastic (+MP). Me (0.25; 0.75), and Mann–Whitney U test. Bold red text—statistically significant differences.

Group	Tumor Nodes Number	Total Area of Tumor Nodes, mm^2^	Tumor Node Diameter, mm
–MP	6 (4; 8)	21.6 (8.7; 26.1)	1.8 (1.3; 2.1)
+MP	12 (11; 16)	38.2 (29.0; 43.6)	1.6 (1.3; 2.0)
*p*-level	** 0.002 **	** 0.018 **	0.9

**Table 2 ijms-26-11511-t002:** Prevalence of pathological changes in the distal colon of mice with AOM/DSS-induced colitis-associated colorectal cancer that did not receive microplastics (–MP) and received microplastics (+MP). % of distal colon section length; Me (0.25; 0.75); Mann–Whitney U test. Bold red text—statistically significant differences.

Group	Tumor	Ulcer	Inflammation	All
–MP	2.9 (1.2; 8.7)	5.9 (4.6; 7.2)	57.2 (53.5; 60.1)	73.4 (62.6; 79.2)
+MP	33.3 (19.2; 45.2)	4 (3.3; 4.5)	58.2 (46.2; 60.2)	98.9 (93.1; 96.4)
*p*-level	** 0.009 **	0.18	0.8	** 0.003 **

**Table 3 ijms-26-11511-t003:** mRNA expression of genes encoding claudins, mucins, proapoptotic factor *Bax*, and proliferation marker *Mki67* relative to expression of beta-actin mRNA in the wall of the medial colon of mice with AOM/DSS-induced colitis-associated colorectal cancer that did not receive microplastics (–MP) and received microplastics (+MP); Me (0.25; 0.75); Mann–Whitney U test.

Genes	–MP	+MP	*p*-Level
*Muc1*	0 (0; 0.2)	0 (0; 0)	0.45
*Muc3*	88 (17; 147)	218 (66; 560)	0.33
*Muc13*	8.6 (0.7; 31.5)	2 (1.7; 6.2)	0.70
*Cldn2*	0.01 (0; 0.22)	0 (0.13)	0.94
*Cldn4*	5.5 (1.2; 15.9)	23.2 (3.7; 56.3)	0.23
*Cldn7*	7 (0.4; 14.3)	8.3 (3.3; 11.3)	0.76
*Bax*	9.5 (0; 40.3)	15.9 (4.2; 21.5)	0.85
*Mki67*	0.2 (0; 18.2)	4.3 (1.3; 35.5)	0.42

**Table 4 ijms-26-11511-t004:** The serum cytokine levels in mice with AOM/DSS-induced colitis-associated colorectal cancer that did not receive microplastics (–MP) and received microplastics (+MP); Me (0.25; 0.75); Mann–Whitney U test.

Cytokines	–MP	+MP	*p*-Level
IL-6	65.3 (62.2; 93.8)	73.9 (69.0; 74.7)	0.72
IL-10	21.9 (19.4; 32.1)	20.6 (19.8; 24.4)	0.81
TNF-α	27.0 (26.8; 30.0)	31.2 (26.4; 33.1)	0.41
IL-1β	36.1 (32.9; 39.0)	36.4 (29.7; 55.2)	0.81

**Table 5 ijms-26-11511-t005:** Primer sequences.

Genes		Primer Sequences
*Actb*	For:	CCTGCCACCCAGCACAAT
	Rev:	GGGCCGGACTCGTCATAC
*Cldn2*	For:	TGCGACACACAGCACAGGCATCAC
	Rev:	TCAGGAACCAGCGGCGAGTAG
*Cldn4*	For:	TCGTGGGTGCTCTGGGGATGCT
	Rev:	GCGGATGACGTTGTGAGCGGTC
*Cldn7*	For:	GCCTTGGTAGCATGTTCCTGGA
	Rev:	GGTACGCAGCTTTGCTTTCACTG
*Muc1*	For:	GGTGACCACTTCTGCCAACT
	Rev:	TCCTTCTGAGAGCCACCACT
*Muc3*	For:	TGTTCAGCTTTACTGTGTTTCAA
	Rev:	TTGCATGTCTCCTCAGGATT
*Muc13*	For:	AGCATGTCCCAGCTTTCTCA
	Rev:	CCATTTGCTGCCTGAGGA
*Bax*	For:	CATGGACTGGAGAAGGGACT
	Rev:	ACCCCATTCTTCCTGATGC
*Mki67*	For:	AGGCGAAGTGGAGCTTCTGA
	Rev:	GCTGCTGCTTCTCCTTCACTG

## Data Availability

The data that support the findings of this study are available on request from the corresponding author.

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
