# Peer review of "Microplastics’ Impact on the Development of AOM/DSS-Induced Colitis-Associated Colorectal Cancer in Mice"

_ijms, 2025, doi:10.3390/ijms262311511_

Round 1

Reviewer 1 Report

Comments and Suggestions for Authors

This study by Natalia Zolotova et al. focused on the effects of plastic microparticles on the development of colitis-associated colorectal cancer in a chemically induced murine model with AOM/DSS. In general, is good, particularly the figures, tables and references. Unfortunately, the text is weaker and needs attention.

  1. The assertion "Animals receiving MPs demonstrated more pronounced inflammatory infiltration in both tumors and non-tumors tissue" should be supported by the determination of pro-inflammatory cytokines in tumor tissue, which the authors should consider determining by PCR the transcripts of TNF-alpha, IL-17, and IL-6, at least.
  2. There are some grammatical, typographical and English errors throughout the text.

In the abstract sectio; the wording regarding the experimental groups is confusing in the abstract section; please rewrite it. Always use the same abbreviation when referring to plastic microparticles, as using MPs, MP, or PS hinders smooth reading.

  1. In the introduction section there are paragraphs that require the reference that supports the observations, for example: in the paragraph "Although MP exposure did not cause pathological changes in the colon, MP-induced oxidative stress, local...." lines 67-70.

In the paragraphs “Chronic inflammation in IBD leads to the formation of a microenvironment development, enriched…”, lines 83--88.

  1. Authors need to ensure that all abbreviations are defined the first time, e.g. PS is similar to MPs or MP?
  2. Check for typos throughout the text, e.g., spaces before parentheses on line 100.

The manuscript, in its current form, does not do justice to either the field of study or the quality of work carried out in the author's laboratory. Greater attention to detail is needed; that is, the English in the manuscript needs to be corrected, as do some inaccurate statements. A rigorous and critical review can easily address all the deficiencies mentioned.

Author Response

Dear reviewer, we thank you for your careful review of our manuscript. We have expanded the introduction and discussion, added additional chapter entitled "Limitations of the Study and Further Research Objectives," improved the English, corrected typos, and removed an extra column in Table 2.

 «This study by Natalia Zolotova et al. focused on the effects of plastic microparticles on the development of colitis-associated colorectal cancer in a chemically induced murine model with AOM/DSS. In general, is good, particularly the figures, tables and references. Unfortunately, the text is weaker and needs attention» - Thank you for your comments. We've improved the article's text.

«1. The assertion "Animals receiving MPs demonstrated more pronounced inflammatory infiltration in both tumors and non-tumors tissue" should be supported by the determination of pro-inflammatory cytokines in tumor tissue, which the authors should consider determining by PCR the transcripts of TNF-alpha, IL-17, and IL-6, at least» - By the term "inflammatory infiltration" we meat the cell number increase in lamina propria. Cellular infiltration is demonstrated in Fig. 5 A-D. We did not use specific markers of immune cells, but based on morphological features, a significant number of inflammatory cells were detected in the lamina propria cell composition: 1) lymphocytes - cells with a round dark nucleus and a thin rim of cytoplasm; 2) plasma cells - cells with a nucleus resembling a "wheel with spokes", next to the nucleus - a large light eosinophilic area of ​​​​the cytoplasm (Golgi apparatus); 3) neutrophils - cells with a segmented nucleus. Furthermore, DSS is known to be a proinflammatory agent. intestinal IL-6, IFN-γ, TNF-α, and IL-17A concentration This AOM/DSS-induced CAC is typically characterized by an increase in proinflammatory cytokines in the colon, such as IL-6, IFN-γ, TNF-α, IL-17A (Giner E et al., 2016, doi: 10.1002/mnfr.201500605). Therefore, the cellular infiltration of the laminae propria we observed is clearly an inflammatory infiltration.

The impact of microplastics on cytokine production in the colon in colorectal cancer is an interesting and important area for the further research.

Unfortunately, we are unable to study cytokine expression in the colon in our current experiment. We only have cDNA libraries for medial colon homogenates, without separating tumor and non-tumor tissue. Our study showed that tumor and non-tumor tissue differ significantly. Furthermore, according to Dzhalilova et al., (2024) (doi: 10.3390/ijms25147801), cytokine expression in tumor and non-tumor colon tissue differs significantly in mice with AOM/DSS-induced CAC. Therefore, to study cytokine gene expression, samples must be collected correctly, which requires a complete repeat of the experiment. The AOM/DSS-induced CAC modeling alone takes 12 weeks, and the journal only allows 10 days for manuscript revision.

We've added this task on analyzing colonic cytokine production to the new discussion chapter "Limitations of the study and further research objectives":

“Local markers of inflammation. We assessed the severity of inflammation using two parameters: cell density in the lamina propria and serum cytokine levels. Although we found increased immune cell infiltration into the intestinal wall following MP exposure, systemic cytokine levels did not differ. Therefore, it is worth examining the effects of MP on local levels of pro- and anti-inflammatory cytokines in the intestinal wall at the level of mRNA expression and protein content. According to most literature, in healthy mice, colon MP consumption led to an increase in the proinflammatory cytokines TNFα, IL-1β, and IL-6 levels, as well as their mRNA expression [19]. In mice with acute and chronic DSS-induced colitis after MP espouse demonstrated higher colon mRNA expression of pro-inflammatory cytokine genes (Tnfa, Il1b, Il6, Il17a, Il22, Tgfb), lower expression of anti-inflammatory cytokine Il10 and increased colonic level of proinflammatory cytokines TNF-α, IL-1β and IL-6 [26–28,86]. According to Dzhalilova et al. [68] in mice with AOM/DSS-induced CAC, cytokine expression in tumor and non-tumor colon tissue differs significantly, so the study must be conducted separately for different tissue types. In addition, we determined the severity of inflammatory infiltration based on the number of all nuclei in the lamina propria. In addition to inflammatory immune cells (proinflammatory macrophages, lymphocytes, plasma cells, and neutrophils), the lamina propria also contains fibroblasts, fibrocytes, anti-inflammatory macrophages, and regulatory T lymphocytes. Migrating tumor cells may also be presented in the tumor area. Therefore, it is also worth assessing the effect of MP on the different immune cell populations content.”

«2. There are some grammatical, typographical and English errors throughout the text». – An english-speaking researcher  has checked and corrected English in the text.

«In the abstract sectio; the wording regarding the experimental groups is confusing in the abstract section; please rewrite it. Always use the same abbreviation when referring to plastic microparticles, as using MPs, MP, or PS hinders smooth reading.» - The term «microplastics» is usually used as a plural noun, so the abbreviation for the plural noun "s" added: "MPs". In some sentences, according to grammar rules, the plural form of a noun cannot be used, so we wrote "MP". This spelling of the word is used in most articles. Microplastics are small particles of any plastic. However, according Xie et al., (2022), the ratio of immune cells in the colon mucosa depend on the type of MP: PS>PVC> PET> PE> PP> control. Therefore, when describing specific studies, we tried to indicate the chemical composition of microplastics. Most oftenly, microplastics are made of polystyrene (PS) or polyethylene (PE) are used in experiments on mice. We have changed the description of the experiment in the "abstract":

“The AOM/DSS-induced CAC model was reproduced in 2 groups of adult male C56BL/6 mice. One of these groups received MP (5 μm polystyrene microbeads) with drinking water at a dose of 1.48 mg/kg/day throughout the experiment (12 weeks), another untreated with MP.”

«3. In the introduction section there are paragraphs that require the reference that supports the observations, for example: in the paragraph "Although MP exposure did not cause pathological changes in the colon, MP-induced oxidative stress, local...." lines 67-70.» - Changed the phrase:

“Although exposure to MPs did not induce observable pathological alterations in the colon [19], we assume that MP-induced oxidative stress, local proinflammatory reactions and barrier dysfunction may exacerbate the intestinal inflammation caused by other factors.”

«In the paragraphs “Chronic inflammation in IBD leads to the formation of a microenvironment development, enriched…”, lines 83--88» - Changed the phrase:

“Chronic inflammation in IBD leads to the microenvironment development, enriched with immune cells that produce proinflammatory cytokines and growth factors and simultaneously increase reactive oxygen species local levels that cause oxidative stress [31].”

«4. Authors need to ensure that all abbreviations are defined the first time, e.g. PS is similar to MPs or MP?» - MP and MPs are the singular and plural forms of microplastics, respectively. The abbreviation's explanation was in the text. PS is the generally accepted designation for polystyrene; the abbreviation's explanation has been added.

«5. Check for typos throughout the text, e.g., spaces before parentheses on line 100 - Thank you for your comment. We have corrected the typos in the text.

«The manuscript, in its current form, does not do justice to either the field of study or the quality of work carried out in the author's laboratory. Greater attention to detail is needed; that is, the English in the manuscript needs to be corrected, as do some inaccurate statements. A rigorous and critical review can easily address all the deficiencies mentioned.» - Thank you for your feedback. We've revised the article, improved the English, corrected typos, and changed inaccurate phrases.

Reviewer 2 Report

Comments and Suggestions for Authors

My comments are as follows:

  1. The use of uniform, pristine 5-μm polystyrene spheres in distilled water does not represent real-world microplastic exposure, which typically involves mixed polymers, irregular shapes, and surface weathering. There is no physicochemical characterization of the particles in suspension, and the actual internal dose in the animals was not measured. Without confirming microplastic accumulation in tissues or feces, the link between exposure and observed effects remains indirect.
  2. Molecular analyses were performed on the medial colon, whereas most tumors developed distally. This likely diluted any local transcriptional effects of MPs. The conclusion that MPs did not influence mucin, claudin, or apoptosis-related gene expression is therefore not well supported. Sampling should focus on the distal colon and distinguish between tumor, peritumoral, and non-tumor areas.
  3. Each group contained only seven mice at the endpoint, which is a small number for histological variability. Multiple comparisons were handled with Bonferroni correction, but many findings are reported as “trends,” which may overstate weak or underpowered effects. A power analysis or effect size reporting is needed to substantiate statistical claims.
  4. Tumor quantification was limited to counts and surface area measurements. There was no grading of dysplasia or adenocarcinoma, nor immunohistochemical validation of proliferation or apoptosis markers. More detailed pathology (Ki-67, β-catenin, cleaved caspase-3) would substantiate the claim that MPs promote cancer rather than benign hyperplasia.
  5. Although the discussion attributes tumor promotion to enhanced inflammation, direct evidence is limited to morphologic infiltration and serum cytokines, which were unchanged. Measurement of cytokines or immune cell markers in colonic tissue, rather than serum, would better demonstrate local inflammatory effects.
  6. The finding that more tumors contained elevated chromogranin A-positive enteroendocrine cells in the MP group is interesting, but the functional significance is unclear. The link between increased enteroendocrine cells and tumor aggressiveness is speculative without correlation to tumor size or histological grade.
  7. Only male mice were studied. Given possible sex-related differences in immune and barrier responses, results cannot be generalized. The use of distilled water instead of standard facility water could also alter baseline physiology. The cause of the one death per group should be stated to rule out MP toxicity unrelated to carcinogenesis.
  8. The statement that MPs “may be risk factors in humans” overreaches the data. This study demonstrates an effect in a severe chemical model of inflammation and cancer in mice, which may not reflect human exposure levels or routes. The conclusion should be toned down to emphasize experimental evidence rather than direct human risk.

Author Response

Dear reviewer, we sincerely thank you for your careful review of our manuscript. We have expanded the introduction and discussion, added a chapter entitled "Limitations of the Study and Further Research Objectives," improved the English, corrected typos, and removed an extra column in Table 2.

«1. The use of uniform, pristine 5-μm polystyrene spheres in distilled water does not represent real-world microplastic exposure, which typically involves mixed polymers, irregular shapes, and surface weathering.» - We’ve added discussion of this question in section "Limitations of the study and further research objectives":

“MP particles shape, size and type. MP particles detected in human colorectal tumors vary considerably according to chemical structure (PP, PE, PA, PC, PET, PS, PVC, PU, ​​PMMA, etc.), shape (fiber, fragment, film) and size (1µm – 1.6 mm) [37–43]. However, experimental conditions must be standardized, controlled, and reproducible. Therefore, experimental models of MP exposure generally use only one type of spherical microparticle of a single standard size for each experiment [39,40,45–47]. We also used one type of particle: 5 µm polystyrene microspheres. However, there is evidence that the effects of magnetic fields depend on the size, shape, type of particles, and their charge. The ratio of immune cells in the colon mucosa depends on the type of MP: PS>PVC> PET> PE> PP> control [78]. Modified nanoparticles PS-NH2 and PS-COOH easier enter into Caco-2 cells and demonstrate stronger toxicity in mice in comparison to nonmodified PS [79]. According to some data obtained, smaller particles can easily cross the intestinal barrier, accumulate in larger quantities in internal organs and cause more pronounced damage [80,81], but according to other data [82], PS particles of 5 μm in size cause more pronounced pro-inflammatory changes in the colon of mice than particles of 0.2 and 1 μm in diameter. Studies where a mixture of different MPs was used are rare [18,83]. When two different size particles are used together, their bioaccumulation changes [18]. The effects of different MPs on the development of CRC should be studied in the future.”

«There is no physicochemical characterization of the particles in suspension» - We did not study the microplastic particles themselves and their properties in suspension, as our team does not include necessary specialists or the necessary equipment. We only have the information provided in the product description on the supplier's website: Quality Level 100, form aqueous suspension, crosslinking 0 % cross-linked, concentration 10% (solids), particle size 5 μm std dev <0.1 μm, coeff var <2%, density 1.05 g/cm3, storage temp 2-8°C (https://www.sigmaaldrich.com/RU/en/product/sial/79633). This information was added to the section "Limitations of the study and further research objectives".

«The actual internal dose in the animals was not measured. Without confirming microplastic accumulation in tissues or feces, the link between exposure and observed effects remains indirect.» - unfortunately we do not have the technical capacity to evaluate MP accumulation. We added discussion of this question in section "Limitations of the study and further research objectives":

“MPs bioaccumulation. We did not evaluate MP accumulation in mouse tissues and feces. However, other authors conducted bioaccumulation studies on the MPs we used (5-μm polystyrene microspheres). According to Liang et al. [18] after 24 h exposure this MP predominantly accumulated in the mice colon. Deng et al. [85] revealed that the particle concentration in liver, kidney and gut tissues reached optimum state within 14 days of the exposure onset, and after 4-week exposure MP colon concentration was 1.4 mg/g. In mice with acute colitis MP abdomen accumulation significantly increased compared to healthy animals [25].”

«2. Molecular analyses were performed on the medial colon, whereas most tumors developed distally. This likely diluted any local transcriptional effects of MPs. The conclusion that MPs did not influence mucin, claudin, or apoptosis-related gene expression is therefore not well supported. Sampling should focus on the distal colon and distinguish between tumor, peritumoral, and non-tumor areas.» - We do not claim that there are no differences in gene expression; we just report that unfortunately were not able to detect any changes in expression. We acknowledge that sampling was unsuccessful and cite this as an explanation to detect differences. Perhaps we shouldn't have published this result, however we decided to warn other researchers about suchmistakes. This problem was added to the section "Limitations of the study and further research objectives":

“Colon tissue sampling. Pathological changes in the colon in the AOM/DSS-induced CAC model were unevenly distributed. Most tumor nodules were detected in the distal colon, so it is necessary to collect material from the distal colon for analysis. Furthermore, tumor and non-tumor tissue differ significantly, so the study should be conducted separately for these tissue types. We likely did not detect the effect of MP on gene expression in the colon because we sampled the medial colon and did not separate the tissue into tumor and non-tumor tissue. That was a significant limitation of our study. However, the mouse colon is small, making it impossible to collect material from the same colonic region simultaneously for morphometric histology and PCR analysis of gene expression, much less to additionally assess protein levels in the intestinal wall. For molecular biology studies, it is advisable to use additional, separate groups of animals.”

Results in abstract corrected: “We did not find any MP effects on the claudins, mucins, proapoptotic factor Bax and proliferation marker Mki67 genes expression in medial colon”

«3. Each group contained only seven mice at the endpoint, which is a small number for histological variability. Multiple comparisons were handled with Bonferroni correction, but many findings are reported as “trends,” which may overstate weak or underpowered effects. A power analysis or effect size reporting is needed to substantiate statistical claims.» - We added discussion of this question in section "Limitations of the study and further research objectives":

“Experimental sample size. Literary data on the effect of MPs on tumor development are still quite sporadic and contradictory. The experimental groups size in published articles are small: from 4 to 10 mice per group [39,40,45,46]. Therefore, we used a small number of experimental animals (n=7). According to the power analysis (two independent study group, continuous, effect size mean, enrollment ratio=1, type I/II-error rate, Alpha=0.05, Power=80%), sample size n=7 is sufficient for parameters such as tumor nodes number and total area of tumor nodes on macrophotos, all pathological changes prevalence in histological sections, lamina propria cell number in tumor and non-tumor tissue, goblet cell highly sulfated mucin content in non-tumor tissue. For tumor prevalence and goblet cell highly sulfated mucin content in tumor tissue, n=8 mice are required, and for goblet cell volume fraction and neutral mucin content, 11-18 animals’ number per group is required (Table S1).”

Tumor and non-tumor tissue differ significantly, and it is not fully correct to combine them into a single group. Therefore, when analyzing histological parameters, the number of comparison groups was increased to four. When comparing more than two groups, it is necessary to introduce a correction for multiple comparisons; we chose correction Bonferroni. Correction for multiple comparisons reduces the risk of finding differences where they do not exist, but at the same time increases the risk of losing differences where they do exist.  Our study was conducted on a small sample of animals. Therefore, to ensure that possible differences are not overlooked, we discussed trends. A trend is not a definitive difference between groups, but rather a guide for future research on which parameters to focus on.

«4. Tumor quantification was limited to counts and surface area measurements. There was no grading of dysplasia or adenocarcinoma, nor immunohistochemical validation of proliferation or apoptosis markers. More detailed pathology (Ki-67, β-catenin, cleaved caspase-3) would substantiate the claim that MPs promote cancer rather than benign hyperplasia.» - We planned to conduct an immunohistochemistry study using antibodies to Ki67, caspase 3, and cytokeratins. However, due to sanctions regimen in our country, we having significant difficultiess with  procuring imported antibodies, and the necessary antibodies are not produced domestically. The delivery of the antibodies we ordered took six months, and the reagents arrived non-functional, likely due to improper storage conditions during transportation. This is not the first time we have encountered this problem and can not find a solution. We would very much like to conduct the study you suggested, but we do not have the necessary reagents, and purchasing new ones will take at least six months, while the journal only allows 10 days to finalize the article. We added discussion of this question in section "Limitations of the study and further research objectives":

“Markers of tumor malignancy and progression. All the tumors we identified exhibited an abnormal growth pattern, consisting of multiple glands lined with proliferating atypical cells with hyperchromatic nuclei of varying shapes and heights. Goblet cells number and mucus production in the tumors were significantly reduced, indicating impaired epithelial cell differentiation. The tumor stroma consisted of connective tissue infiltrated with neutrophils, lymphocytes, and histiocytes. None of the cases demonstrated tumor invasion into the submucosa. As both tissue and cellular atypia were observed, we classified the tumors we observed to adenocarcinomas. However, for an accurate assessment of the differentiation progression and malignancy of the tumor, an immunohistochemical study of a number of markers should be carried out: proliferation marker Ki-67, a component of the Wnt signaling pathway involved in the regulation of proliferation and migration - β-catenin, apoptosis inductor caspase-3, tumor suppressor p53, proteins of intermediate filaments of epithelial cells cytokeratins [87].”

«5. Although the discussion attributes tumor promotion to enhanced inflammation, direct evidence is limited to morphologic infiltration and serum cytokines, which were unchanged. Measurement of cytokines or immune cell markers in colonic tissue, rather than serum, would better demonstrate local inflammatory effects.» - We don't have the tissue samples necessary to study cytokines and inflammatory markers in the intestinal wall. To conduct such a study, we need to fully repeate the experiment. Reproducing the CRC model alone takes 12 weeks, and the journal gives us 10 days to correct the article. However, we have added a discussion of this problem to Section «Limitations of the study and further research objectives»:

“Local markers of inflammation. We assessed the severity of inflammation using two parameters: cell density in the lamina propria and serum cytokine levels. Although we found increased immune cell infiltration into the intestinal wall following MP exposure, systemic cytokine levels did not differ. Therefore, it is worth examining the effects of MP on local levels of pro- and anti-inflammatory cytokines in the intestinal wall at the level of mRNA expression and protein content. According to most literature, in healthy mice, colon MP consumption led to an increase in the proinflammatory cytokines TNFα, IL-1β, and IL-6 levels, as well as their mRNA expression [19]. In mice with acute and chronic DSS-induced colitis after MP espouse demonstrated higher colon mRNA expression of pro-inflammatory cytokine genes (Tnfa, Il1b, Il6, Il17a, Il22, Tgfb), lower expression of anti-inflammatory cytokine Il10 and increased colonic level of proinflammatory cytokines TNF-α, IL-1β and IL-6 [26–28,86]. According to Dzhalilova et al. [68] in mice with AOM/DSS-induced CAC, cytokine expression in tumor and non-tumor colon tissue differs significantly, so the study must be conducted separately for different tissue types. In addition, we determined the severity of inflammatory infiltration based on the number of all nuclei in the lamina propria. In addition to inflammatory immune cells (proinflammatory macrophages, lymphocytes, plasma cells, and neutrophils), the lamina propria also contains fibroblasts, fibrocytes, anti-inflammatory macrophages, and regulatory T lymphocytes. Migrating tumor cells may also be presented in the tumor area. Therefore, it is also worth assessing the effect of MP on the different immune cell populations content.”

«6. The finding that more tumors contained elevated chromogranin A-positive enteroendocrine cells in the MP group is interesting, but the functional significance is unclear. The link between increased enteroendocrine cells and tumor aggressiveness is speculative without correlation to tumor size or histological grade.» – We expanded the discussion of enteroendocrine cells:

“EECs are cells of the gastrointestinal tract that secrete hormones that regulate gastrointestinal motility, nutrient sensing, and glucose homeostasis. They also play a role in immune responses and can influence appetite and satiety. In the colon, EECs comprise approximately 1% of epithelial cells. Chromogranin A is a general marker for EECs [55]. According to the single-cell transcriptome analysis [56], in the mouse colon EEC are represented by the following 4 cell types: 1) 48% Enterochromaffin (EC) cell, secrete serotonin (5-TH); 2) 35% L-cell, secrete Glucagon-like peptide-1 and Peptide YY; 3) 11% D-cell, secrete Somatostatin; 4) 7% Insulin-like peptide 5 (Insl5)-producing cells. Thus, the predominant hormone of the colon is serotonin. In the intestine, serotonin stimulates peristalsis, mucus secretion, chloride, and bicarbonate, inhibits water absorption, causes vasodilation, and modulates immune responses by attracting immune cells and exerting a pro-inflammatory effect [57]. Our previous studies demonstrated that in healthy mice exposed to MPs, the number of EECs and immune cells in the colon increases, and goblet cell volume fraction decreases [22,23]. We suggest that MPs mechanically irritate the colonic wall, causing increased secretion and production of serotonin. Serotonin induces hypersecretion and emptying of colonic cells, and moreover attracts immune cells to the intestinal mucosa. These are adaptive responses aimed at flushing MPs from the body and strengthening local immune defenses. In mice with CAC, the number of EECs in non-tumor tissue did not change when exposed to MP, but in tumors it varied significantly in both MP-exposed and non-MP-exposed mice. However, in mice consuming MP, the number of tumor nodules with increased EEC content was higher. It could be suggested that EEC hormones may modulate tumor progression by stimulating epithelial cell proliferation, vascularization and recruiting immune cells. However, research on the connection between the presence of enterocytes in colorectal carcinomas and prognosis is inconclusive [58]. According to Chen et al. [59], high EEC number and chromogranin A level in CRC is positively correlated with invasion and lymph node metastasis.”

«7. Only male mice were studied. Given possible sex-related differences in immune and barrier responses, results cannot be generalized.» - We have added discussion of this question to the section «Limitations of the study and further research objectives»:

“Sex differences. Sex hormones and sex chromosome genes significantly influence the immune system, leading to sex differences in the incidence and the infectious, inflammatory, autoimmune, and neoplastic diseases severity [93,94]. In particular, CRC morbidity and mortality are higher in men than in women. Moreover, left-sided CRC is predominant in men, while women are more often diagnosed with the more aggressive right-sided CRC [95]. In the AOM/DSS-induced CAC model, the number of tumor nodules and their sizes are greater among males than among females [96]. Therefore, it is impossible to include both males and females in the same sample. The female hormonal cycle leads to cyclical changes in immune function [97]. In women, many chronic diseases experience worsening symptoms before or during menstruation [98]. In acute ozone exposure in the lungs of female mice in the follicular phase (proestrus, estrus), a significantly more pronounced proinflammatory reaction is observed than in the luteal phase (metestrus, diestrus) [99]. Therefore, studies on females must take into account the stability and phase of the estrous cycle, which makes the work more labor-intensive. Furthermore, fluctuations in sex hormones increase the variability of responses, so studies on females, especially with small sample sizes, risk missing effects. Therefore, most studies are first conducted on males, and the results are then confirmed and supplemented with sex-specific data in studies on females. Undoubtedly, future research on the MPs effects on the colorectal cancer development in females is an important task.”

«The use of distilled water instead of standard facility water could also alter baseline physiology.» – Microplastics was detected in tap and bottled water (Gambino et  al., 2022, doi:10.3390/ijerph19095283). Furthermore, the rate of microplastic particle aggregation is proportional to the ion content of the water (Al Harraq & Bharti, 2021,  doi:10.1021/acsenvironau.1c00016). To ensure that the microplastic suspension remains stable and prevents the ingress of foreign microplastic particles, many studies use deionized water (miliQ) for microplastic suspension. Tap water can't be used to prepare microplastic suspensions, and giving mice deionized water for 12 weeks is also undesirable. Therefore, we chose distilled water. The mouse feed we used contains all the necessary macro- and microelements, so using distilled water had no negative effect. The experiment on the effect of MP on colorectal cancer development is part of a larger study. There was a control group of mice that received distilled water for 12 weeks. We did not observe any physiological changes in them (Zolotova et al.,  2025, doi:10.3390/toxics13080701).

«The cause of the one death per group should be stated to rule out MP toxicity unrelated to carcinogenesis.» - An autopsy was not performed on the dead mice, so we were not able to indicate the exact cause of death. But the deaths were not related to microplastics. Оne animal in the group not exposed to microplastics also died. We had groups of healthy mice and mice with chronic colitis that were also exposed to microplastics for 12 weeks, and all animals survived. When modeling colitis at high doses of DSS and/or long-term exposure, animal mortality observes. Death is associated with a severe inflammatory process in the colon, leading to the development of systemic inflammatory reactions and dehydration (Chassaing et al., 2014, doi:10.1002/0471142735.im1525s104; Jiang Xet al., 2023, doi:10.17219/acem/156647).

«8. The statement that MPs “may be risk factors in humans” overreaches the data. This study demonstrates an effect in a severe chemical model of inflammation and cancer in mice, which may not reflect human exposure levels or routes. The conclusion should be toned down to emphasize experimental evidence rather than direct human risk.» – changed the conclusion to “MPs may represent a potential environmental cofactor contributing to CAC risk”

Reviewer 3 Report

Comments and Suggestions for Authors

Weaknesses and Points for Improvement

  1. Sample size: Only eight mice per group (with one death per group) may limit statistical power; consider discussing this limitation.

  2. Localization of analysis: qRT-PCR was performed on medial colon, while most tumors occurred in the distal region—this is acknowledged but could be emphasized more clearly in the Discussion as a design limitation.

  3. Gene expression results: Lack of observed molecular differences could be due to sampling site or overshadowing by AOM/DSS effects; further explanation or alternative assays (e.g., protein-level validation) would strengthen the conclusions.

  4. Systemic effects: The study reports no cytokine changes; discussion could benefit from more speculation on local vs systemic inflammation mechanisms.

  5. Figures: Some micrographs (Figures 3–7) could benefit from higher resolution or annotated labels to improve readability.

  6. Language and style: While generally good, a few grammatical errors and awkward phrasings should be edited for smoother English and consistency (e.g., “we did not find any MP effects of on the expression…”).

  7. The is incorrect order in manuscript sections - results before materials

Author Response

Dear reviewer, we sincerely thank you for your careful review of our manuscript. We tried to expand the introduction and discussion, added a chapter entitled "Limitations of the Study and Further Research Objectives," improved the English, corrected typos, and removed an extra column in Table 2.

«1. Sample size: Only eight mice per group (with one death per group) may limit statistical power; consider discussing this limitation.» - We have added discussion of this question to the section «Limitations of the study and further research objectives»:

“Experimental sample size. Literary data on the effect of MPs on tumor development are still quite sporadic and contradictory. The experimental groups size in published articles are small: from 4 to 10 mice per group [39,40,45,46]. Therefore, we used a small number of experimental animals (n=7). According to the power analysis (two independent study group, continuous, effect size mean, enrollment ratio=1, type I/II-error rate, Alpha=0.05, Power=80%), sample size n=7 is sufficient for parameters such as tumor nodes number and total area of tumor nodes on macrophotos, all pathological changes prevalence in histological sections, lamina propria cell number in tumor and non-tumor tissue, goblet cell highly sulfated mucin content in non-tumor tissue. For tumor prevalence and goblet cell highly sulfated mucin content in tumor tissue, n=8 mice are required, and for goblet cell volume fraction and neutral mucin content, 11-18 animals’ number per group is required (Table S1).”

«2. Localization of analysis: qRT-PCR was performed on medial colon, while most tumors occurred in the distal region—this is acknowledged but could be emphasized more clearly in the Discussion as a design limitation.» - We have added discussion of this question to the section «Limitations of the study and further research objectives»:

“Colon tissue sampling. Pathological changes in the colon in the AOM/DSS-induced CAC model were unevenly distributed. Most tumor nodules were detected in the distal colon, so it is necessary to collect material from the distal colon for analysis. Furthermore, tumor and non-tumor tissue differ significantly, so the study should be conducted separately for these tissue types. We likely did not detect the effect of MP on gene expression in the colon because we sampled the medial colon and did not separate the tissue into tumor and non-tumor tissue. That was a significant limitation of our study. However, the mouse colon is small, making it impossible to collect material from the same colonic region simultaneously for morphometric histology and PCR analysis of gene expression, much less to additionally assess protein levels in the intestinal wall. For molecular biology studies, it is advisable to use additional, separate groups of animals.”

«3. Gene expression results: Lack of observed molecular differences could be due to sampling site or overshadowing by AOM/DSS effects; further explanation or alternative assays (e.g., protein-level validation) would strengthen the conclusions.» - We don't have colon samples for protein analysis. To conduct a full study, we need to repeat the entire experiment. Modeling CRC alone takes 12 weeks, and the journal only allows 10 days to revise the article.

«4. Systemic effects: The study reports no cytokine changes; discussion could benefit from more speculation on local vs systemic inflammation mechanisms.» - We expanded the discussion of systemic inflammatory responses:

“In addition to local symptoms, CRC also causes a range of systemic manifestations, including a systemic inflammatory response. This chronic, dysregulated inflammatory state significantly impacts tumor progression and is a marker of poor prognosis [69]. In comparison to the healed individuals, CRC patients demonstrated from 3.2 to 4.4-fold higher concentrations of the proinflammatory cytokines, including IL-1β, IL-6, and TNF-α. On the contrary, the level of the anti-inflammatory cytokine IL-10 was reduced [70]. Comparing the cytokine levels in this study with our previous analysis of cytokine levels in healthy animals [23], both MP-treated and MP-untreated mice with CAC had approximately 1.5-fold elevated TNF-alpha levels, while other cytokine levels were normal. We revealed no effect of MPs on serum levels of IL-1β, IL-6, TNF-α, and IL-10 in mice with CAC. It is likely that the MPs effects are too weak in the context of the severe pathological process caused by AOM/DSS. The exact mechanism by which DSS causes colitis is not fully understood. It was demonstrated that DSS disrupted the mucus layer, caused epithelial cell death, and stimulated the innate immune response, triggering the secretion of proinflammatory cytokines and chemokines. Damage to the epithelial barrier leads to the translocation of dietary and bacterial antigens to the mucosa and the development of an inflammatory reaction in the intestinal wall. Increased intestinal barrier permeability, especially in ulcer areas, led to the release of luminal antigens into the bloodstream, which triggers systemic inflammatory reactions [71]. MP also caused an increase in colon permeability: a number of studies demonstrated the increase in intestinal permeability for fluorochrome-labeled dextran with a molecular weight of 4 kDa and 70 kDa when consuming MP [18,72–77]. But the effects of MP are much weaker than those of DSS.”

«5. Figures: Some micrographs (Figures 3–7) could benefit from higher resolution or annotated labels to improve readability.» - All illustrations have a resolution of 300 pixels per inch and a width of 20 cm. Small illustrations in the text of the article in the electronic version of the journal contain links to full-size illustrations. All photos contain arrows or lines highlighting significant elements. All symbols are explained in the captions.

«6. Language and style: While generally good, a few grammatical errors and awkward phrasings should be edited for smoother English and consistency (e.g., “we did not find any MP effects of on the expression…”).» - A researcher with a good knowledge of English corrected the text.

«7. The is incorrect order in manuscript sections - results before materials» – This section sequence is a requirement of the IJMS journal: «Research manuscript sections: Introduction, Results, Discussion, Materials and Methods, Conclusions (optional)» [https://www.mdpi.com/journal/ijms/instructions]. We formatted the article strictly according to the journal template.

Reviewer 4 Report

Comments and Suggestions for Authors

This manuscript presents an original and well-structured experimental study addressing the effect of polystyrene microplastic (PS-MP) exposure on the development of colitis-associated colorectal cancer (CAC) using an AOM/DSS-induced mouse model. The topic is timely, relevant, and of growing global concern, given increasing evidence of microplastic exposure and its potential health risks. The study is competently designed and executed, with a clear rationale and methodology, well-presented data, and a balanced discussion. The work contributes meaningful insights into how MPs may exacerbate intestinal inflammation and carcinogenesis, thereby providing a foundation for future investigations into environmental factors influencing colorectal cancer risk.

  1. Although the manuscript is understandable, the English could be polished for smoother flow and readability (e.g., “we find only 5 works” → “only five previous studies have addressed this issue”). Minor grammatical errors (articles, verb tenses) and repetitive phrasing should be corrected during copyediting.
  2. The study convincingly shows a tumor-promoting effect of MPs but lacks mechanistic assays to delineate pathways (e.g., oxidative stress markers, ROS quantification, DNA damage). Including immunostaining for proliferation (Ki67) and apoptosis (cleaved caspase-3) at the tissue level could reinforce the findings.

  3. The authors mention that gene expression analyses were performed on the medial colon, while most tumors developed in the distal colon. This is a critical limitation that should be highlighted more strongly in the discussion and addressed in future experiments.
  4. The absence of changes in serum cytokines may reflect the dominance of AOM/DSS-induced systemic inflammation, masking subtle MP effects. As sugestion, measuring local cytokine expression in colon tissue (rather than serum alone) might have provided a more sensitive assessment.

  5. Only eight mice per group (with one death in each) is a small sample size for variability inherent in AOM/DSS models. A power analysis or mention of variability limitations would strengthen the credibility of statistical claims.
  6. The conclusion could be refined to avoid overgeneralization (e.g., “MPs may likely be risk factors for CAC in humans” could be softened to “MPs may represent a potential environmental cofactor contributing to CAC risk”).
  7. In the introduction, it would be important for the authors to address the differences between the cellular and tissue-level impacts of microplastics and nanoplastics. In addition, mentioning the global impacts of plastic pollution would also strengthen the context of the study.
  8. Although the authors present the methodology clearly and objectively, there appears to be a potential source of bias. The microplastic particles were added to the drinking water and homogenized several times a day; however, how can this process be faithfully reproduced in practice? Additionally, providing all intestinal and nodule images as supplementary material would greatly enhance the study’s credibility. The authors should address this point in the discussion.
  9. I suggest that the primer sequences be presented in a table format to improve clarity and readability. This would allow readers to easily identify the genes, primer sequences, and corresponding references in a more organized manner.
  10. I also recommend that the authors include a graphical abstract. A well-designed visual summary would help readers quickly grasp the main findings and overall significance of the study, thereby increasing its visibility and impact.

Author Response

Dear reviewer, we thank you for your careful review of our manuscript. We have expanded the introduction and discussion, added a chapter entitled "Limitations of the Study and Further Research Objectives," improved the English, corrected typos, and removed an extra column in Table 2.

«This manuscript presents an original and well-structured experimental study addressing the effect of polystyrene microplastic (PS-MP) exposure on the development of colitis-associated colorectal cancer (CAC) using an AOM/DSS-induced mouse model. The topic is timely, relevant, and of growing global concern, given increasing evidence of microplastic exposure and its potential health risks. The study is competently designed and executed, with a clear rationale and methodology, well-presented data, and a balanced discussion. The work contributes meaningful insights into how MPs may exacerbate intestinal inflammation and carcinogenesis, thereby providing a foundation for future investigations into environmental factors influencing colorectal cancer risk.» - Thank you for your high appreciation of our work

«1. Although the manuscript is understandable, the English could be polished for smoother flow and readability (e.g., “we find only 5 works” → “only five previous studies have addressed this issue”). Minor grammatical errors (articles, verb tenses) and repetitive phrasing should be corrected during copyediting.» - An english-speaking researcher with a good knowledge of English has checked and corrected English in the text

«2. The study convincingly shows a tumor-promoting effect of MPs but lacks mechanistic assays to delineate pathways (e.g., oxidative stress markers, ROS quantification, DNA damage). Including immunostaining for proliferation (Ki67) and apoptosis (cleaved caspase-3) at the tissue level could reinforce the findings.» – Investigating the mechanisms of MP effect was not our goal. To date, only three studies have been published evaluating the effect of microplastics on the development of AOM/DSS-induced CAC. Two of these studies revealed that MP stimulated tumor growth, while the third suppressed it. Moreover, all three studies used nanoplastic particles, which more easily cross the intestinal barrier than microplastics and therefore, in most cases, have more pronounced effects. Therefore, we were uncertain whether MP would have any effects on tumors in our experiment. However, we have included a discussion of the mechanisms and the need for immunohistochemical tumor analysis in the chapter "Limitations of the study and further research objectives":

“Markers of tumor malignancy and progression. All the tumors we identified exhibited an abnormal growth pattern, consisting of multiple glands lined with proliferating atypical cells with hyperchromatic nuclei of varying shapes and heights. Goblet cells number and mucus production in the tumors were significantly reduced, indicating impaired epithelial cell differentiation. The tumor stroma consisted of connective tissue infiltrated with neutrophils, lymphocytes, and histiocytes. None of the cases demonstrated tumor invasion into the submucosa. As both tissue and cellular atypia were observed, we classified the tumors we observed to adenocarcinomas. However, for an accurate assessment of the differentiation progression and malignancy of the tumor, an immunohistochemical study of a number of markers should be carried out: proliferation marker Ki-67, a component of the Wnt signaling pathway involved in the regulation of proliferation and migration - β-catenin, apoptosis inductor caspase-3, tumor suppressor p53, proteins of intermediate filaments of epithelial cells cytokeratins [87].

Mechanisms of MP effect. Our aim was to reveal whether MP provides an effect on colorectal tumor growth in vivo; we did not investigate the mechanisms involved. In vitro studies demonstrated that MP induces the reactive oxygen species development and the oxidative DNA damage, which leads to decreased cell viability and the initiation of autophagy or apoptosis [88]. In vivo, in the healthy mice colon, the main negative effects of MPs are increased intestinal permeability, activation of oxidative stress and proinflammatory reactions, and changes in the microbiota composition [19]. Moreover, suppression of the gut microbiota with antibiotics mitigates the effects of MPs [89–92]. According to Tian et al. [46], in mice with AOM/DSS-induced CAC, exposure to 20 nm polystyrene nanoparticles in the colon resulted in suppression of fatty acid metabolism, activation of lipid peroxidation, increased ROS levels, significantly more severe DNA damage, increased abundance of colitogenic bacterium Allobaculum in the intestinal microflora, and decreased postbiotic bacterium Lactobacillus. Further studies, concerning the MPs effect mechanisms in tumor development are an important scientific goal.”

«3. The authors mention that gene expression analyses were performed on the medial colon, while most tumors developed in the distal colon. This is a critical limitation that should be highlighted more strongly in the discussion and addressed in future experiments.» - We have added discussion of this question to the section «Limitations of the study and further research objectives»:

“Colon tissue sampling. Pathological changes in the colon in the AOM/DSS-induced CAC model were unevenly distributed. Most tumor nodules were detected in the distal colon, so it is necessary to collect material from the distal colon for analysis. Furthermore, tumor and non-tumor tissue differ significantly, so the study should be conducted separately for these tissue types. We likely did not detect the effect of MP on gene expression in the colon because we sampled the medial colon and did not separate the tissue into tumor and non-tumor tissue. That was a significant limitation of our study. However, the mouse colon is small, making it impossible to collect material from the same colonic region simultaneously for morphometric histology and PCR analysis of gene expression, much less to additionally assess protein levels in the intestinal wall. For molecular biology studies, it is advisable to use additional, separate groups of animals.”

«4. The absence of changes in serum cytokines may reflect the dominance of AOM/DSS-induced systemic inflammation, masking subtle MP effects. As sugestion, measuring local cytokine expression in colon tissue (rather than serum alone) might have provided a more sensitive assessment.» - Since tumor and non-tumor tissue differ significantly, local inflammatory markers must be assessed separately for these tissue types. We do not have the necessary intestinal samples for analysis. To conduct a study of local inflammatory markers, the experiment must be completely repeated. Modeling colorectal cancer alone takes 12 weeks, and the journal allows 10 days for revision of the article. However, we have added a discussion of this issue in the section "Limitations of the study and further research objectives"

“Local markers of inflammation. We assessed the severity of inflammation using two parameters: cell density in the lamina propria and serum cytokine levels. Although we found increased immune cell infiltration into the intestinal wall following MP exposure, systemic cytokine levels did not differ. Therefore, it is worth examining the effects of MP on local levels of pro- and anti-inflammatory cytokines in the intestinal wall at the level of mRNA expression and protein content. According to most literature, in healthy mice, colon MP consumption led to an increase in the proinflammatory cytokines TNFα, IL-1β, and IL-6 levels, as well as their mRNA expression [19]. In mice with acute and chronic DSS-induced colitis after MP espouse demonstrated higher colon mRNA expression of pro-inflammatory cytokine genes (Tnfa, Il1b, Il6, Il17a, Il22, Tgfb), lower expression of anti-inflammatory cytokine Il10 and increased colonic level of proinflammatory cytokines TNF-α, IL-1β and IL-6 [26–28,86]. According to Dzhalilova et al. [68] in mice with AOM/DSS-induced CAC, cytokine expression in tumor and non-tumor colon tissue differs significantly, so the study must be conducted separately for different tissue types. In addition, we determined the severity of inflammatory infiltration based on the number of all nuclei in the lamina propria. In addition to inflammatory immune cells (proinflammatory macrophages, lymphocytes, plasma cells, and neutrophils), the lamina propria also contains fibroblasts, fibrocytes, anti-inflammatory macrophages, and regulatory T lymphocytes. Migrating tumor cells may also be presented in the tumor area. Therefore, it is also worth assessing the effect of MP on the different immune cell populations content.”

«5. Only eight mice per group (with one death in each) is a small sample size for variability inherent in AOM/DSS models. A power analysis or mention of variability limitations would strengthen the credibility of statistical claims.» - We have added discussion of this question to the section «Limitations of the study and further research objectives»:

“Experimental sample size. Literary data on the effect of MPs on tumor development are still quite sporadic and contradictory. The experimental groups size in published articles are small: from 4 to 10 mice per group [39,40,45,46]. Therefore, we used a small number of experimental animals (n=7). According to the power analysis (two independent study group, continuous, effect size mean, enrollment ratio=1, type I/II-error rate, Alpha=0.05, Power=80%), sample size n=7 is sufficient for parameters such as tumor nodes number and total area of tumor nodes on macrophotos, all pathological changes prevalence in histological sections, lamina propria cell number in tumor and non-tumor tissue, goblet cell highly sulfated mucin content in non-tumor tissue. For tumor prevalence and goblet cell highly sulfated mucin content in tumor tissue, n=8 mice are required, and for goblet cell volume fraction and neutral mucin content, 11-18 animals’ number per group is required (Table S1).”

«6. The conclusion could be refined to avoid overgeneralization (e.g., “MPs may likely be risk factors for CAC in humans” could be softened to “MPs may represent a potential environmental cofactor contributing to CAC risk”).» - Thanks for the advice. We've changed the conclusion based on your recommendation.

«7. In the introduction, it would be important for the authors to address the differences between the cellular and tissue-level impacts of microplastics and nanoplastics.» - We have expanded the introduction:

“All of these studies included plastic nanoparticles. It was assumed that NPs are potentially more dangerous than MP. Due to their smaller size, nanoparticles can easily cross biological barriers, such as the gut-blood barrier, and translocate to organs and tissues throughout the body. Moreover, nanoparticles can penetrate into cells, disrupting intracellular functions. NPs have much higher surface area-to-volume ratio in comparison to MPs. This larger surface area leads to greater adsorption of harmful substances and bacteria and toxic plastic components release [48,49]. However, experimental data on the MPs and NPs harm ratio is controversial [48].”

«In addition, mentioning the global impacts of plastic pollution would also strengthen the context of the study» - Sorry, we didn't understand what you mean by "global impacts of plastic pollution." The article provides general information about plastic pollution and the prevalence of microplastics:

“Plastic industrial production began around the 1950s, it's rising steadily with projections indicating it will continue to rise significantly by 2050. Only about 9% of all plastic waste is being recycled and 19% is being incinerated, with the remainder ending up in landfills (50%) or mismanaged waste systems that pollute the environment (22%). Plastic pollution is considered as a global environmental problem. The greatest concern is caused by plastic particles smaller than 5 mm – microplastics (MPs)…», «MPs are categorized by their minute size, low mass, and environmental persistence. The properties allow MPs to be spread by the wind and accumulate in soils and water, which leads to the growth of habitat pollution. MPs particles were found in urban air, agricultural soil, rivers, oceans, drinking water, and human food. Therefore, the MPs’ impact on human health is an important issue [1–3]. Evaluations of human MP consumption are mostly indirect and based on MP content in water and food estimates. According to various evaluation, human MP intake is from 0.2 to 10.2 mg/kg/day [4,5] or from 1.5 to 1.1 × 106 particles/kg/day [6–10].”

The article concerns the  medical issue. We provided a detailed review of microplastic pollution in our previous work (Zolotova, N.; Kosyreva, A.; Dzhalilova, D.; Makarova, O.; Fokichev, N. Harmful Effects of the Microplastic Pollution on Animal Health: A Literature Review. Peerj 2022, 10, e13503, doi:10.7717/peerj.13503.).

«8. Although the authors present the methodology clearly and objectively, there appears to be a potential source of bias. The microplastic particles were added to the drinking water and homogenized several times a day; however, how can this process be faithfully reproduced in practice? Additionally, providing all intestinal and nodule images as supplementary material would greatly enhance the study’s credibility. The authors should address this point in the discussion.» - we have added a discussion of this issue in the section "Limitations of the study and further research objectives":

“MP intake dose. There are two main methods for modeling oral MP intake: 1) administering MPs via a stomach tube; 2) adding MPs to drinking bowls [1]. In the first case, the dose can be precisely controlled. However, humans are exposed to MPs through water and food continuously, throughout the day. With intragastric administration, mice receive the entire daily dose at one point: the peak dose of MPs is very high, followed by no further MP intake for the next 24 hours. Furthermore, the effects of MPs on healthy mice are relatively weak, in comparison to nonspecific stress reactions [19,84]. Daily insertion of a tube into the stomach is a strong stressor for mice. The effects of the procedure itself may overwhelm the effects of MPs. In the second option, when adding MP to the water bowls, it is technically impossible to control the exact liquid volume consumed by each mouse. In addition, the suspension of MP particles is unstable; the particles gradually settle, but the sedimentation rate is slow. Calculated using Stokes' Law settling velocity for 5 µm spherical polystyrene particles in distilled water is 7.03 x 10-7 m/s or approximately 6 cm per day. Although this model does not allow to control MP doses precisely, it significantly better matches with the dynamics of human MP consumption and does not provide stress for animals.”

«Additionally, providing all intestinal and nodule images as supplementary material would greatly enhance the study’s credibility.»- We added Figure S1: Macrophotos of all mice colons. Scans of the entire distal region are very large files, each nearly 100 MB in size. Even with archiving, the folder containing all the images is over 1 GB in size—too much for supplementary storage (The journal allows supplementary materials no larger than 60 MB). We can send these files to you personally.

«9. I suggest that the primer sequences be presented in a table format to improve clarity and readability. This would allow readers to easily identify the genes, primer sequences, and corresponding references in a more organized manner.» - The primer sequences were converted into a tabular format

«10. I also recommend that the authors include a graphical abstract. A well-designed visual summary would help readers quickly grasp the main findings and overall significance of the study, thereby increasing its visibility and impact.» - We made a graphical abstract.

Round 2

Reviewer 2 Report

Comments and Suggestions for Authors

All my comments and concerns have been addressed. 

Reviewer 3 Report

Comments and Suggestions for Authors

  All corrections significantly improved the quality of paper. However, the quantity of studied groups is still   too small for statistical analysis. All limitations of study are well described. 

Comments on the Quality of English Language

It is fine.